# Detectability of the Impacts of Ozone Depleting Substances and Greenhouse Gases upon Stratospheric Ozone Accounting for Nonlinearities in Historical Forcings

Justin Bandoro[1], Susan Solomon[1], Benjamin D. Santer[2], Douglas E. Kinnison[3], Michael J. Mills[3]

[1]Department of Earth, Atmospheric, and Planetary Sciences, Massachusetts Institute of Technology, Cambridge, MA 02139
[2]Program for Climate Model Diagnosis and Intercomparison (PCMDI), Lawrence Livermore National Laboratory, Livermore, CA 94550
[3]Atmospheric Chemistry Observations and Modeling Laboratory, National Center for Atmospheric Research, Boulder, CO 80307

*Correspondence to*: Justin Bandoro (jbandoro@mit.edu)

**Abstract.** We perform a formal attribution study of upper and lower stratospheric ozone changes using observations together with simulations from the Whole Atmosphere Community Climate Model. Historical model simulations were used to estimate the zonal-mean response patterns ("fingerprints") to combined forcing by ozone depleting substances (ODS) and well-mixed greenhouse gases (GHG), as well as to the individual forcing by each factor. Trends in the similarity between the searched-for fingerprints and homogenized observations of stratospheric ozone were compared to trends in pattern similarity between the fingerprints and the internally and naturally generated variability inferred from long control runs. This yields estimated signal-to-noise (S/N) ratios for each of the three fingerprints (ODS, GHG, and ODS+GHG). In both the upper stratosphere (defined in this paper as 1 to 10 hPa) and lower stratosphere (40 to 100 hPa), the spatial fingerprints of the ODS+GHG and ODS only patterns were consistently detectable not only during the era of maximum ozone depletion, but also throughout the observational record (1984-2016). We also develop a fingerprint attribution method to account for forcings whose time evolutions are markedly nonlinear over the observational record. When the nonlinearity of the time evolution of the ODS and ODS+GHG signals are accounted for, we find that the S/N ratios obtained with the stratospheric ODS and ODS+GHG fingerprints are enhanced relative to standard linear trend analysis. Use of the nonlinear signal detection method also reduces the detection time - the estimate of the date at which ODS and GHG impacts on ozone can be formally identified. Furthermore, by explicitly considering nonlinear signal evolution, the complete observational record can be used in the S/N analysis, without applying piece-wise linear regression and introducing arbitrary break points. The GHG-driven fingerprint of ozone changes was not statistically identifiable in either the upper or lower stratospheric SWOOSH data, irrespective of the signal detection method used. In the WACCM simulations of future climate change, the GHG signal is statistically identifiable between 2020-2030. Our findings demonstrate the importance of continued stratospheric ozone monitoring to improve estimates of the contributions of ODS and GHG forcing to global changes in stratospheric ozone.

## 1 Introduction

Climate change detection and attribution ("D&A") studies seek to identify and formally quantify an anthropogenic component of change in observed climate data. Formal identification of an anthropogenic climate change "fingerprint" has been successfully achieved with observations of atmosphere and ocean temperatures, sea level, ocean acidity, various components of the water cycle and the cryosphere, and certain climate extremes (Bindoff et al., 2013). To date, however, few formal D&A methods have been applied in studies involving stratospheric ozone (see Gillett et al. 2011 for one exception to this). There is evidence that stratospheric ozone is transitioning from an era of widespread and readily detectable depletion (linked to changes in anthropogenic chlorofluorocarbons) to an era characterized by early signs of recovery or healing (Solomon et al., 2016). Our motivation for this work is to determine whether formal D&A methods can provide a more confident and quantitative attribution of ozone depletion and recovery signals.

Global changes in the physical climate system are driven by both internal variability and external influences (Hegerl et al., 2007; Karl et al., 2006). Internal variability is generated through complex interactions of the coupled atmosphere–ocean system. External influences include human-caused changes in well-mixed greenhouse gases (GHG), ozone depleting substances (ODS), and other radiative forcing agents, as well as natural fluctuations in solar irradiance and volcanic aerosols. Past D&A studies have found that each of these external influences has a unique "fingerprint" in the detailed zonal-mean latitude/altitude pattern of temperature change (Hansen et al., 2005; Karoly et al., 1994; Santer et al., 1996a; Tett et al., 1996; Thorne et al., 2002; Vinnikov et al., 1996). The use of such profiles of atmospheric temperature change has proved particularly useful in separating human, solar, and volcanic influences on climate, and in discriminating between externally forced signals and internal variability. In this study, we use the latitude/altitude patterns of both upper and lower stratospheric ozone change in response to individual and combined anthropogenic forcings to understand the relative detectability of ODS and GHG signals in observations.

Stratospheric ozone depletion has been a significant international concern since it was first recognized as a consequence of anthropogenic emissions of ODS (Molina and Rowland, 1974). Following implementation of the Montreal Protocol, there was a decline in emissions of ODS and a consequent decrease in halogen-containing compounds in the stratosphere. In the 2014 World Meteorological Organization (WMO) Scientific Assessment of Ozone Depletion (WMO, 2014), a major topic of interest was the statistical significance of ozone trends over the last decade, and the extent to which "ozone recovery" is taking place. Assessing the statistical significance of observed ozone trends does not yield definitive information on the causes of trends. In addition to determining whether observed negative or positive trends in ozone are unusually large or small relative to model-based estimates of unforced trends in ozone, it is also important to understand and quantify the contributions of different climate forcings to the observed ozone changes, and to assess whether the agreement between an externally forced fingerprint and observations could have been obtained by natural causes alone.

There is a clear scientific consensus that man-made chlorofluorocarbons were the dominant driver of global stratospheric ozone decline from the 1970s to 2000s (e.g., Solomon, 1999 and references therein). It is also well established that increasing greenhouse gases contribute to a EE of the middle to upper stratosphere (Boville, 1986; Fels, 1980). Colder temperatures result in a slowing down of gas-phase catalytic reactions that destroy ozone in the upper stratosphere, which in the absence of any other changes, leads to ozone increases. Thus, both decreasing ODS and increasing GHG concentrations can act to increase upper stratospheric ozone. Understanding the extent to which GHG increases may confound the attribution of ozone changes due to ODS is essential to the identification of ozone recovery at these altitudes (see WMO, 2014).

Another potential confounding factor is change in the Brewer-Dobson circulation (BDC), the meridional overturning circulation in the middle atmosphere that transports trace gases from the tropics to the poles (Brewer, 1949; Dobson, 1956). Previous studies have found that a strengthening of the BDC results in lower ozone concentrations in the tropical lower stratosphere and an increase in concentrations in the extratropics, and modeling studies have suggested that increases in GHG concentrations lead to a strengthening of the BDC (Eyring et al., 2010; Fleming et al., 2011; Garcia et al., 2008; Gillett et al., 2011; Oman et al., 2010). In contrast, a recent study by Polvani et al. (2017) found that trends in ODS, and not in GHG levels, have been the primary driver of trends in tropical upwelling. This complicates the problem of attributing ozone changes that are primarily BDC-related, since both GHG and ODS forcing may be implicated in driving changes in the BDC.

Another forcing that can affect the concentration of ozone in the stratosphere is the solar cycle. The total solar irradiance changes by about 0.1% during the 11 year solar cycle; however, UV radiation can change by about 4-8% (Lean, 2000). Stratospheric ozone changes during the solar cycle have been well documented (e.g., Merkel et al., 2011) with the largest percentage change in ozone between solar maximum and minimum occurring between 30-45 km. Stratospheric ozone is also influenced by internal variability, primarily via the quasi-biennial oscillation (QBO). The effect of the QBO on upper stratospheric ozone has been shown to be small compared to its larger signal in interannual ozone variability in the lower and middle stratosphere (Hasebe, 1994; Zawodny and McCormick, 1991). These natural and internal changes must also be considered in attributing ozone changes to specific causes.

Most D&A studies have been focused on the attribution of changes in the climate system to anthropogenic GHG forcing, which has been increasing linearly since the early 20[th] century. In contrast, ODS forcing has had a unique nonlinear time evolution because of the implementation of the Montreal Protocol. For analyses spanning both the depletion and "ozone recovery" periods, purely linear trends are inadequate for capturing the more complex nonlinear evolution of ozone change in certain parts of the stratosphere (as we will show later). Many ozone trend studies address this nonlinear behavior by performing piecewise linear regression with a break point around 1997 (Bourassa et al., 2014; Chehade et al., 2014; Jones et al., 2009;

Kyrölä et al., 2013; Laine et al., 2014). The slopes of the piecewise trends are not constrained by physical and chemical considerations, and are typically arbitrarily chosen to enhance the slopes for each of the eras. Another approach is to quantify changes over the entire record by regression to a nonlinear proxy, such as the halogen loading in the stratosphere, described by the equivalent effective stratospheric chlorine (EESC, see e.g. Newchurch, 2003). Many studies have employed EESC-based regression to quantify trends in ozone (e.g. Langematz et al., 2016; Stolarski et al., 2006a; Wohltmann et al., 2007). This approach utilizes the entire observational record, with no arbitrary start or end points. While EESC-based regression offers a number of advantages for the detection of nonlinear ODS-driven ozone changes, it also has certain disadvantages (Kuttippurath et al., 2015). In section 4, we use EESC-based regression to address the problem of nonlinear ODS evolution over the full observational record. Our D&A approach also accounts for the much smaller temporal changes in the GHG forcing time series.

To the best of the authors knowledge, only one previous study has considered the formal attribution of anthropogenic emissions to observed stratospheric ozone changes: Gillett et al. (2011). Their investigation examined zonal-mean Solar Backscatter Ultraviolet (SBUV) ozone measurements (Mclinden et al., 2009) over the 27-year period from 1979 to 2005 in the middle and upper stratosphere (50-1 hPa). Model results were taken from simulations performed as part of the Chemistry-Climate Model Validation activity (CCMVal, Eyring et al. 2010). Gillett et al. analyzed multi-model simulations performed with greenhouse gases only, combined anthropogenic factors only, and combined anthropogenic and natural external forcings. The ozone response to changes in ODS and the response to changes in natural external forcings were estimated through subtraction of simulations. Gillett et al. (2011) used a standard space-time optimal regression methodology for signal detection and attribution (see, e.g., Allen and Tett, 1999). In this approach, the observations are modeled as a linear sum of simulated responses ("fingerprints") to individual forcings, with each response scaled by an estimated regression coefficient (expressing the strength of the space-time response pattern in observations). A regression coefficient significantly greater than zero indicates a detectable response to the forcing, and a coefficient close to unity signifies that simulated and observed responses are similar in magnitude (attribution). The underlying premise here is that the observations can be well-represented by a linear combination of the input model signal response fields and an additive noise term due to internal climate variability. It is also assumed that the response patterns to different individual forcings are statistically distinct (i.e., are separable), and that the sum of the individual responses is equivalent to the response obtained when all forcings are varied simultaneously.

Gillett et al. (2011) were unable to separate the individual ODS and GHG responses in the SBUV ozone data, but found a clear combined anthropogenic signal that was consistent with observations. In their study, the authors hypothesized that the difficulty in separating the individual ODS and GHG responses was due to multiple factors: the limited ensemble size, and the degeneracy between the patterns of stratospheric ozone response to ODS and GHG forcing. Since the optimal regression methodology combines the spatial and temporal response into a single space-time vector, it was not possible to determine whether the degeneracy between the ODS and GHG response patterns was primarily due to spatial similarity or to similarity in temporal evolution.

We use a different approach here to understand and quantify the relative contributions (and detectability) of the ozone responses to ODS and GHG forcing. Rather than combining spatial pattern and time evolution information in a single vector, we use pattern correlations to assess the time evolution of the spatial similarity between time-invariant fingerprints and time sequences of: 1) observed ozone patterns; and 2) model-based estimates of the natural variability of ozone. Such methods rely on some form of spatial covariance statistic (e.g., Santer et al., 1993, 1995), and may involve rotation of the fingerprint in a low-noise direction in order to optimize signal-to-noise ratios (Hasselmann, 1993; Hegerl et al., 1996) . The conventional strategy in this method is to search for a long-term, positive trend in the pattern correspondence statistic, which would indicate an increasing expression of the searched-for signal in the observations. The underlying assumption here is that the spatial pattern of the fingerprint does not change markedly as a function of time, which is a reasonable assumption for historical GHG forcing, but is probably reasonable for ODS forcing (see Solomon et al., 2016).

Our study differs from that of Gillett et al. (2011) in a number of ways. In addition to using a different D&A method and a longer ozone dataset, we are also addressing different statistical questions. First, we seek to determine whether the observed changes in stratospheric ozone are unusual relative to estimates of both internal climate variability and total natural variability (Santer et al., 2013). Second, we consider the time evolution of signal-to-noise (S/N) ratios, both for the individual fingerprints of ODS-forced and GHG-forced ozone changes, as well as for the ozone fingerprint arising from combined ODS+GHG forcing. This allows us to determine the detection time – the time at which S/N exceeds and remains consistently above some stipulated significance level. Third, we calculate S/N ratios separately for ozone changes in the upper and lower stratosphere rather than for a single region only (the 1 to 50 hPa region considered by Gillett et al.). The reasons for this decision are explained below. Fourth, because we do not combine spatial and temporal information, the spatial aspects of S/N behavior are easier to evaluate (e.g., in terms of the pattern similarity between the searched-for fingerprints and the observations).

There are several disadvantages of our selected approach. Unlike Gillett et al. (2011), who performed a multi-model analysis, we rely on a single model only, and are unable to evaluate the sensitivity of our results to possible errors in the model-based estimates of internal variability and the response to external forcing (North et al., 1998). Additionally, since we do not explicitly incorporate time evolution information in the searched-for fingerprints, pronounced differences in the time evolution of the ozone responses to ODS and GHG forcing cannot be used in the separation of these responses (at least not in the "linear trend" representation of these signals.

As noted above, we perform separate D&A analyses for ozone changes in the upper and lower stratosphere. Our premise is that different dominant processes control the changes in ozone concentrations in these two regions . In the lower stratosphere, particularly in the tropics and extratropics, ozone concentrations are affected by direct transport, both through interannual variability and via changes in the BDC (Shindell and Grewe, 2002). A further influence on lower stratospheric ozone

concentrations occurs by means of heterogeneous chemistry on the surfaces of polar stratospheric clouds and volcanic aerosols; the heterogeneous reactions are extremely sensitive to temperature changes (Solomon, 1999). The ozone chemical lifetime is on the order of weeks in the lower stratosphere in the Antarctic, and on the order of months at mid-latitudes.

In the upper stratosphere, the processes controlling ozone abundance are quite different; the photochemical lifetime of ozone is on the order of hours, so that direct transport of ozone is not important. The local concentrations of upper stratospheric ozone are determined by gas-phase chemistry, which is well understood and well replicated in current chemistry-climate models (Austin et al., 2002). Transport of other greenhouse gases (such as methane) can indirectly affect ozone concentrations in the upper stratosphere (Solomon and Garcia, 1984). The anticipated response from GHG forcing radiatively cools the upper
stratosphere uniformly with latitude (Aquila et al., 2016), increasing ozone concentrations, and potentially impacting atmospheric circulation through the BDC. A strengthening of the BDC would result in lower stratospheric ozone decreases in the tropics and increases in the extratropics.

Based on this understanding from atmospheric physics and chemistry, we expect that different forcings should yield distinctly
different latitudinal and vertical patterns of change. Below, we examine in detail the use of these patterns for attribution of the causes of ozone changes in the upper and lower stratosphere.

## 2 Model Simulations and Observed Ozone Datasets

The model used for all simulations analyzed in this study is version 1 of the Community Earth System Model, with version 4 of the Whole Atmosphere Community Climate Model as the atmospheric component (CESM1(WACCM); see Marsh et al.
2013). WACCM is a coupled chemistry-climate model that extends from the Earth's surface to the lower thermosphere. WACCM's representation of heterogeneous chemistry has been shown to be in broad agreement with observations of polar ozone and related chemical species (Solomon et al., 2015). A wide range of experiments were performed with WACCM:

*i)* Simulations that follow the experimental design of the Chemistry-Climate Model Intercomparison (CCMI) REFC1
experiment (Morgenstern et al., 2017), with volcanic aerosols prescribed according to CCMI (Arfeuille et al., 2013). These simulations include coupled chemistry and dynamics (Marsh et al., 2013) with sea surface temperatures prescribed according to observations for the period 1955-01 to 2014-12. Concentrations of ozone depleting substances and greenhouse gases vary over time, as specified in the REFC1 scenario. A set of 5 ensemble members was available, each with identical forcings but starting from slightly differing initial conditions. We refer to these simulations subsequently as ALL1.

*ii)* Free-running simulations that follow the same REFC1 scenario as in ALL1, with the same atmospheric coupled chemistry and dynamics, but including an interactive fully-coupled ocean component. These simulations begin in 1960-01 and extend

through 2099-12. One set of runs includes both historical time-varying greenhouse gases and ozone depleting substance concentrations (referred to herein as ALL2), while another keeps GHG concentrations fixed at their 1960 conditions but allows the evolution of time-varying ODS concentrations (referred to as FIXED GHG1960). A third set complements the second with ODS concentrations fixed at 1960 conditions but with time-varying GHG concentrations (FIXED ODS1960). Each of the
aforementioned sets of experiments contains 3 ensemble members that were performed with slightly different initial conditions. FIXED GHG1960 and FIXED ODS1960 follow the convention of the CCMI REFC2 experiments (Eyring et al., 2013). It should be noted that the time-varying concentrations of ODSs in FIXED GHG1960 are radiatively active.

All sets of simulations in *i)* and *ii)* include identical historical external forcings from solar insolation changes and prescribed
CCMI volcanic aerosol evolution. As in many other general circulation models (see Xue et al., 2012), the quasi-biennial oscillation (QBO) is imposed by nudging the tropical stratospheric zonal winds to observed winds. From the sets of fully-coupled simulations, we were able to estimate responses to greenhouse gases alone (referred to herein as GHGonly) and ozone depleting substances (referred to herein as ODSonly) by independently differencing the individual ensemble anomalies from the FIXED GHG1960 and ODS1960 simulations from the ALL2 ensemble mean. In addition, we were able to isolate the
combined response to volcanic aerosols, the solar cycle, and the QBO by differencing the sum of ensemble mean anomalies of the FIXED GHG1960 and ODS1960 simulations from the ensemble anomalies of ALL2. We designate this as the natural historical signal (referred to herein as NAT-h). The underlying assumption for our method of estimating the GHGonly, ODSonly and NAT-h responses is that the stratospheric ozone responses to individual forcings add linearly. Eyring et al. (2010) found this assumption to hold in the stratosphere, but with some departures from linearity in the tropical total column
ozone. In Supplementary Figure S1 we show that, in absence of the NAT-h response, the global-mean sum of the GHG and ODS response in the upper and lower stratospheric ozone are close to the all forcing simulation.

There is reason to suspect that the NAT-h signal may not thoroughly represent the ozone response to volcanic eruptions in certain regions of the lower stratosphere. In the low halogen loading stratospheric conditions of FIXED ODS1960, increases
in mid-stratospheric ozone following a large volcanic eruption are expected as the loss of ozone at these altitudes is dominated by odd nitrogen, due to enhancement of $N_2O_5$ hydrolysis (Tie and Brasseur, 1995). In the enhanced halogen loading conditions of FIXED GHG1960 and ALL2, the mid-stratospheric increase is limited to higher altitudes, and ozone depletion will occur throughout the lower stratosphere. The 40 to 100 hPa region in the lower stratosphere encompasses a part of the vertical altitude range where the opposing effects of reactive nitrogen and halogen chemistry occur. The validity of the assumption that these
opposing effects are additive is unclear. Despite the uncertainty in the simulated ozone response to volcanic forcing, we show subsequently that the NAT-h simulation successfully captures many features of the observed short- and long-term variability in lower stratospheric ozone.

*iii)* Free-running fully coupled simulations of the last millennium (years 850 to 1850), in which only the estimated external forcing from historical solar variability is specified. This simulation (henceforth referred to as NAT) provides insight into the range of fluctuations in stratospheric ozone arising from the combined effects of the solar cycle and internal variability (other than the QBO; there is no nudging to observed stratospheric winds in this simulation).

*iv)* Free-running fully coupled pre-industrial control run of length 200 years. Unlike *iii)*, there are no temporal changes in solar forcing and all variability is intrinsic to the climate system. This is the same experiment used by Marsh et al. (2013) to provide an estimate of internal climate variability alone, and is referred to herein as CTL.

We rely primarily on the Stratospheric Water and Ozone Homogenized (SWOOSH) database (Davis et al., 2016) for observational estimates of stratospheric ozone changes. SWOOSH includes vertically resolved ozone from a subset of the limb profiling satellite instruments operating from 1984 to present day. SWOOSH's ozone product is a gridded monthly-mean zonal-mean time-series of mixing ratios on pressure levels ranging from 1 to 316 hPa. A key aspect of this merged product is that the source records are homogenized to account for inter-satellite biases and to reduce the impact of non-climatic artifacts.

In this study, we use SWOOSH version 2.6 with a latitudinal resolution of 10 degrees. An advantage of the SWOOSH dataset is that the vertical range enables investigation of the full stratosphere from 1984-01 to 2016-12. Tummon et al. (2015) found that SWOOSH ozone concentrations from 1984-2011 were within +/-10% of other datasets throughout much of the stratosphere, with the largest differences in the lower stratosphere.  To test the sensitivity of our D&A results for the upper stratosphere to the choice of observational dataset, we also employ a second ozone dataset that is independent from SWOOSH:

the previously-mentioned SBUV Merged Cohesive (SBUV_CDR) ozone dataset. This is based on the version 8.6 SBUV MOD dataset, and attempts to further reduce the inter-satellite differences. The dataset is produced at the National Oceanic and Atmospheric Administration (NOAA, [ftp://ftp.cpc.ncep.noaa.gov/SBUV_CDR](ftp://ftp.cpc.ncep.noaa.gov/SBUV_CDR)), and was available for the years from 1979-2015 at the time of this study.

For the signal-to-noise analysis, it was necessary to transform the ozone data from both observational grids and the model grid of WACCM to a common grid. The common horizontal grid chosen was SWOOSH's 10 degree latitude zonal-mean grid from 85°S-85°N. Transformation to a relatively coarse-resolution grid reduces the spatial dimensionality of the input datasets, which is of benefit in the estimation of Empirical Orthogonal Functions (EOFs) used later in the fingerprint analysis. Regridding of the vertical coordinates was also performed: the WACCM output vertical levels were transformed to match SWOOSH's

vertical levels (which contains 31 pressure layers, spanning a pressure range from 1 to 316 hPa). Model ozone concentrations were masked at latitude bands and layers where no data was present in SWOOSH. As a sensitivity test, we changed the definition of the upper and lower stratosphere regions used for the D&A analysis, using +/-1 SWOOSH levels on either side of the original vertical ranges. When calculating the fingerprints for the upper and lower stratosphere, we use anomalies weighted by the cosine of latitude, but do not employ vertical pressure weighting. The latter processing choice is reasonable,

since ozone concentrations do not vary by over an order of magnitude within each region and the anomalies are expressed as percent changes.

## 3 Global Mean Ozone Changes

Figures 1 A and B show, separately for the upper and lower stratosphere, the time series of globally averaged annual-mean ozone anomalies for the SWOOSH observations and the five WACCM ensembles (ALL1, ALL2, GHGonly, ODSonly, and NAT-h). Anomalies are expressed in terms of percent changes relative to the first 10 years (1960-1969) of the ALL1 simulation. In the upper stratosphere, there is a steady decline in stratospheric ozone from the 1960s to late 1990s in both the ALL1 and ALL2 historical simulations, which is in agreement with the SWOOSH data. After the late 1990s, the decline slows down, and there is some indication of a recovery of stratospheric ozone. The ODSonly anomalies broadly track the temporal changes in ALL1 and ALL2 but reach larger depletion before beginning to increase, supporting the large body of literature indicating that the large decline was predominantly due to ODS. The difference between the ALL1/ALL2 simulations and ODSonly can be explained by the steady increase in upper stratospheric ozone in GHGonly, consistent with upper stratospheric cooling. This supports the conclusion that upper stratospheric ozone depletion would have been greater during the depletion era in the absence of increases in well-mixed greenhouse gases (WMO, 2014). The NAT-h anomalies show the presence of the 11-year solar cycle in the upper stratosphere, but no overall change in ozone from 1960 to 2016.

In the lower stratosphere (Fig. 1B), ozone changes are characterized by larger interannual variability. There is a dominant 2-3 year cycle that is due to the historically imposed QBO in all ensemble members of ALL1, ALL2, and NAT-h. Although the ODSonly and GHGonly simulations do not contain the natural and QBO response present in the other simulations, we see a large inter-ensemble spread in the lower stratosphere when compared to the upper stratosphere. This is related to dynamical differences between ensemble members as a result of the direct transport of ozone in the lower stratosphere. The ODSonly evolution is similar to that in the upper stratosphere, and broadly tracks the ALL1 and ALL2 runs. For global GHGonly ozone anomalies in the lower stratosphere, there is no significant trend. As noted by Li et al., 2009, in the lower stratosphere, enhanced ozone advection due to a strong BDC results in decreases in the tropical ozone and increases in the extratropical ozone, and in a global mean sense, the lower stratospheric changes cancel out.

It is noteworthy that the historical simulations differ from SWOOSH after 2005, when the global-mean lower stratospheric ozone in SWOOSH does not flatten or begin to turn around, in contrast to the recovery in ALL1 and ALL2. To further examine this divergent behavior, we calculate lower stratosphere ozone changes averaged over the tropics (30°S to 30°N; see Fig. 1C), and over a global domain that excludes the tropics (Fig. 1D). It is clear from this comparison that the post-2005 observation-model difference in global-mean lower stratospheric ozone is largely due to ozone differences in the tropics. This may be partly due to the large observational uncertainty in ozone loss in this region. As noted in the WMO (2014) report, there is

considerable disagreement between observed data sets in terms of ozone changes in the tropical lower stratosphere since 2000. The report states that changes since 2000, "computed from different data sets in the tropical lowermost stratosphere remain an open question". Because of this observational uncertainty in tropical ozone changes (and because of the discrepancy between post-2000 tropical ozone changes in SWOOSH and the ALL1/2 simulations), our subsequent D&A analysis for the lower stratosphere is performed over two domains: a full global domain, and a domain which excludes 30ºS to 30ºN.

To summarize, upper stratospheric ozone in SWOOSH and the ALL1 and ALL2 simulations shows a large decline of 5-8% per decade through the middle 1990s, followed by an increase of 2.5-5%/decade over the last 10 to 15 years. This decline and recovery is in agreement with findings of the WMO (2014) report. In the lower stratosphere, larger interannual variability complicates the identification of long-term ozone changes, and there are post-2005 differences between the historical WACCM model simulations and SWOOSH data that are relevant to the interpretation of the D&A results. SBUV_CDR ozone anomalies in the upper stratosphere are similar to those of SWOOSH (see Supplementary Figure S2).

### 3.1 Variability in WACCM Natural and Control Simulations

The CTL and NAT simulations provide estimates of the amplitude and timescales of natural ozone variability in the WACCM model (see Fig. 2). In the CTL simulation (upper panels), fluctuations in upper and lower stratospheric ozone concentrations arise from internal variability alone. In the last millennium NAT simulation, ozone fluctuations result from both internal variability and from solar forcing (lower panels). Previous work has shown that the solar cycle, ENSO, and the QBO are the main contributory factors to natural variability in stratospheric ozone, affecting ozone concentrations through chemistry and transport mechanisms (Kirgis et al., 2013; Nair et al., 2013). We show below the different effects of these contributory factors in the upper and lower stratosphere.

In the upper stratosphere (left panel of Fig. 2), the ozone variation caused by the 11-year solar cycle is clearly evident in the NAT simulation (see expanded time axis). The magnitude of the internally generated ozone variability in the CTL is smaller than ozone variability arising from the 11-year solar cycle. The NAT-derived estimate of the change in upper stratospheric ozone from solar maximum to solar minimum is approximately 2-4%. This is in agreement with results from previous observational and model-based studies (Lee and Smith, 2003; Newchurch et al., 2003; Randel and Wu, 2007). NAT also shows longer-timescale variability in upper stratospheric ozone concentrations in response to solar insolation changes over multiple decades to centuries. The NAT results highlight the large contribution of solar forcing to ozone variability in the upper stratosphere, and the smaller role played by internally generated variability. The NAT-h simulation (the yellow time series in Fig. 1A) indicates that there is limited influence of the QBO 2-3 year on ozone variability in the upper stratosphere. This is consistent with a previous estimate that the QBO effect on upper stratospheric ozone is less than half that of solar variability over the 11-year solar cycle (Egorova et al., 2004).

In the lower stratosphere (right panel of Fig. 2), the interannual variability of ozone in the CTL is as large as the ozone variability in NAT. In this region of the atmosphere, internal variability in dynamics and transport is the dominant driver of natural fluctuations in ozone – not solar irradiance changes. In fact, the influence of the solar cycle on ozone is difficult to discern in the lower stratosphere (see expanded axis). This is consistent with results from previous studies, which have shown that the solar cycle effect on ozone maximizes around 35 km and weakens in the lower stratosphere (Egorova et al., 2004; Lee and Smith, 2003; Newchurch et al., 2003). The QBO-driven anomalies in NAT-h (Fig. 1B) are of much larger magnitude in the lower stratosphere; QBO-driven ozone anomalies range from 4-8%, in accord with earlier results (Hasebe, 1994; Randel et al., 1996).

Newchurch (2003) found that the large interannual variability in global-mean lower stratospheric ozone cannot be explained by the solar cycle and the QBO only: a significant portion of the variance is related to monthly meteorological variability. Changes in interannual tropical upwelling and lower stratospheric ozone concentrations have been previously linked to the El Niño-Southern Oscillation (ENSO, Bronnimann et al., 2013; Oman et al., 2013). ENSO-induced variations in lower stratospheric ozone were found to exceed 5%. ENSO-related variations in tropical upwelling are partly responsible for modulating the strength of planetary waves, leading to temperature and water vapor variations in the tropical lower stratosphere that can have impacts on the chemistry and transport of ozone (Randel et al., 2009). ENSO signals in ozone are strongest just above the tropical tropopause; at higher levels, they are weaker than the QBO-related ozone signal (Randel and Thompson, 2011).

Supplementary Figure S3 shows the normalized power spectra of global upper and lower stratospheric ozone anomalies for NAT, NAT-h, and CTL. In the upper stratosphere, the 11-year solar cycle is the strongest signal present in the NAT and NAT-h simulations. In the lower stratosphere, the 2-3 QBO signal is strongest in NAT-h (which has an imposed QBO).

By comparing the stratospheric ozone results in Figs. 1 and 2, we can obtain a rough estimate of the relative sizes of anthropogenically-driven and purely natural changes. This simple comparison suggests that the upper stratosphere is the region where anthropogenic signal detection would be most feasible.

**3.2 Observed Versus Model Variability**

Model estimates of internal climate variability are a key component of detection and attribution studies (Allen and Tett, 1999; Santer et al., 2013a). One common strategy is to estimate and remove externally forced climate signals from the observations, and then compare the residual variability with control run internal variability (Hegerl et al., 1996). There are a number of uncertainties in such signal removal strategies (Santer et al., 1996b). Here we use an approach similar to Santer et al., 2013b and directly compare estimates of total variability (arising from both internal processes and natural external factors) in the observations, CTL, NAT, NAT-h, and ALL1 runs. If the model systematically underestimates the amplitude of stratospheric

ozone variability on multi-decadal timescales, the S/N ratios are likely to be inflated. Whether such a systematic error exists is difficult to determine because the global ozone observational records are relatively short. Observations cannot, therefore, provide an unambiguous constraint on model-based estimates of low-frequency ozone changes, but they can provide useful insights into the direction and size of model variability errors.

We investigate these issues with the modeled and observed monthly-mean time series of global-mean upper and lower stratospheric ozone anomalies. After detrending the modeled and observed time series, we isolate long-term variability by applying a band-pass filter with half-power cutoffs at 5 and 20 years to the residuals. We also used a high-pass filter with a half-power point at 3 years to exclude all variability on timescales longer than 5 years. All filtering operations were performed

after detrending the time series with a low-pass filter with a half-power point of 30 years. The key point here is that the modeled and observed ozone data are filtered in exactly the same way. We use $S_{IV}$ ("interannual variability") and $S_{DV}$ ("decadal variability") to denote the temporal standard deviations of the high-pass and band-pass filtered ozone data, respectively.

The $S_{IV}$ and $S_{DV}$ results in Figure 3 were calculated for SWOOSH (396 months), ALL1 (for the 372-month period from 1984

to 2014), NAT (12,000 months), CTL (2,400 months) and NAT-h (2,112 months). NAT (NAT-h) provides information on solar and internal variability in the absence (presence) of variability from the QBO. The CTL variability is solely generated by processes internal to the climate system, and has no contribution from natural external forcing. Fig. 3 shows that these simulations have systematic differences in the amplitude of variability. These differences are manifest on both the sub-3-year and the 5- to 20-year timescales. In the lower stratosphere, the QBO is a dominant component of the variability on timescales

< 3 years: only the ALL1 and NAT-h simulations (both of which have QBO-induced ozone changes) are close to the observed value of $S_{IV}$. NAT-h also includes volcanic variability, which is an important component for $S_{IV}$ in the lower stratosphere. CTL and NAT (which both lack QBO and volcanic-driven ozone changes) underestimate the observed interannual variability. In the upper stratosphere, the observed and modeled $S_{IV}$ all fall within 0.60 - 0.85%.

The decadal variability is of key interest in D&A studies, since it constitutes the background noise against which analysts attempt to identify gradually evolving anthropogenic signals. In the lower stratosphere, the CTL value of $S_{DV}$ is (as expected) lower than in the other three types of simulation. There is no evidence that the simulations most directly comparable to the observations (ALL1 and NAT-h) systematically underestimate the observed values of $S_{DV}$ obtained from the SWOOSH data. Such an underestimate would be concerning: it would spuriously inflate the S/N ratios obtained in the D&A analysis (see

below). In fact, both ALL1 and NAT-h yield values of $S_{DV}$ that are slightly larger than in observations (see Santer et al., 2013a).

In the upper stratosphere, however, all four types of model simulation have values of $S_{DV}$ that are smaller than the observed result. As in the lower stratosphere, CTL (which does not include solar forcing) that has the lowest decadal variability. ALL1, which has most realistic time-varying external forcing, is closest to SWOOSH, but still 11% less than the observed $S_{DV}$ value.

One possible explanation for this difference in upper stratospheric $S_{DV}$ between SWOOSH and ALL1, can be seen in Fig. 1A. In the mid- to late 1980s SWOOSH is prominently higher than ALL1, which would lead to a higher $S_{DV}$ value. There is also considerable observational uncertainty in $S_{DV}$ as different ozone datasets have significantly different representations of the magnitude of the solar cycle (Maycock et al., 2016). Chemistry climate models are also known to have solar cycle variations in ozone that are towards the lower bounds of observational estimates (see Chapter 8.5 of SPARC CCMVAL, 2010).

## 4 Latitude/Altitude Patterns of Ozone Change

Figure 1 shows that in the case of all simulations except GHGonly, a simple least-squares linear fit is not an adequate representation of ozone changes over the entire observational record (1984-2016). The nonlinear behavior of ozone in Fig. 1A occurs because ODS emissions were curtailed through implementation of the Montreal Protocol. As mentioned in the introduction, we address this nonlinearity by representing decadal changes in ozone using an EESC proxy as in Randel and Wu (2007) and Newchurch (2003) (see also WMO 2002, Fioletov and Shepherd, 2005; Stolarski et al., 2006). This proxy isolates ozone changes associated with changes in the amount of ozone depleting chlorine and bromine in the stratosphere. The EESC curve was taken from the NASA Goddard Automailer, which uses the Newman and Daniel (2007) EESC values updated according to WMO (2010). The EESC calculations assume that the mean age of stratospheric air is 5.5 years and age-of-air spectrum width = 2.75 years. Sensitivity to the choice of these two parameters (e.g. varying the MAA from 3-6 years) affected the magnitude but not the significance of the regression coefficients. The EESC fit removes some of the nonlinearity that is manifest in the ozone changes.

Figures 4 A and B show the respective least-squares linear trends in ozone and in EESC over the entire period of observational record (1984-2016). Results are for the ensemble-mean model simulations and the SWOOSH ozone data. The purpose of the comparison is to determine which trend patterns the expect change from individual and combined forcings. We also compare our fingerprints, derived in the next section, to the patterns of long-term change.

We used the method outlined by Trenberth (1984) to determine the statistical significance of the trends. Here and subsequently, the stipulated significance level is 5%. When trends are directly calculated from the ozone data, GHGonly is the only simulation yielding an appreciable region of statistically significant results (Fig. 4A). This is the region where GHGonly shows a 1-2% per decade increase in ozone in the upper stratosphere (1-10 hPa, Fig. 1A) and is the expected region of latitudinally coherent cooling. In GHGonly, there is a non-significant decrease in ozone in the tropical lower stratosphere and a non-significant increase in the extratropics, as expected from an enhanced BDC. WMO (2011, 2014) reported negative ozone trends in the tropical lower stratosphere between 1985-2005, and CCMVal simulations indicate a long-term increase in tropical upwelling and an increased BDC strength. We note, however, that the WMO (2014) analysis of shorter satellite data sets between 2002

and 2012 does not show significant tropical lower stratospheric ozone trends, which may reflect the larger noise associated with substantial decadal variability in the lower stratosphere (see Fig. 3).

Very different results are obtained using the EESC regression (Fig. 4B). As expected, ODS forcing makes the largest contribution to the ozone-change pattern in ALL1 and ALL2. Large regions with significant effects of ODS changes on ozone are evident in ALL1, ALL2 and ODSonly. Each of these simulations shows the familiar pattern of mid-latitude lobes in the upper stratosphere. Another common aspect of ALL1, ALL2, and ODSonly is a lower stratospheric response in the southern hemisphere (SH), with an Antarctic ozone hole that is strong enough to persist in annual anomalies. The SWOOSH ozone data also display the significant two-lobe structure in the upper stratosphere and the lower stratospheric response in the SH polar region. The latter feature, however, extends further equatorward in the ALL1, ALL2, and ODSonly simulations than in the observations. In the historical NAT-h simulation, there are no statistically significant trends in either the ozone or EESC data over the 1984 to 2016 period.

The key conclusion of this section is that the statistical significance of linear trends in stratospheric ozone behavior depends critically on whether trends account for nonlinearities in the ozone forcing. Previous studies with EESC-type regressions have shown a familiar pattern of minima in the upper stratosphere (~35-45km) and the polar lower stratosphere (15-25km), similar to that obtained here in the EESC-based ALL1, ALL2, and ODSonly simulations (Randel and Wu, 2007; Wang et al., 2002). The upper stratospheric changes have a symmetric latitudinal structure, which is the fingerprint of gas-phase chlorine-induced ozone loss. For both ozone and EESC, there are no significant trends in NAT-h over 1984-2016. We conclude from our results that in the WACCM model, ozone changes over 1984 to 2016 are primarily forced by human-caused changes in ODS, and cannot be explained by natural factors alone. Patterns similar to those found for the SWOOSH data (but with slightly larger magnitudes) were obtained using the SBUV_CDR observations (compare Figure 4 and Supplementary Figure S4).

## 5 Fingerprint Estimation

In most applications, the climate change fingerprint is a geographical pattern (Hegerl et al., 1996; Santer et al., 2003), a vertical profile through the atmosphere or ocean (Barnett et al., 2005; Santer et al., 1996b; Tett et al., 1996) or a vector with information on the combined spatial and temporal properties of the signal (Gillett, 2002; Stott et al., 2000; Tett et al., 2002). Here, the fingerprint is a time-invariant latitude-altitude pattern; ozone changes are zonally averaged along latitude bands. The fingerprint provides an estimate of the multi-decadal response to external forcing by combined and individual human forcings. The implicit assumption in this approach is that the spatial pattern of response does not change markedly over time (Santer et al., 2013). We examine the adequacy of this assumption for the specific problem of interest here – the identification of a human-caused fingerprint pattern in the SWOOSH observations. The assumption of timescale-invariance of the fingerprint is tested by defining the fingerprint over different time intervals. We use a standard method (Hasselmann, 1979; Santer et al.,

1995) to determine whether model-predicted patterns of externally forced stratospheric ozone changes can be identified in SWOOSH. Fingerprinting is performed separately for the upper and lower stratosphere.

Let $\chi(i, x, p, t)$ represent the annual-mean percent ozone anomalies at latitude band $x$, pressure $p$, and year $t$ for the $i^{\text{th}}$ ensemble realization for each of the ALL1, ODSonly and GHGonly simulations:

$i = 1, \dots N_r$ (number of ensemble members, ranging from 3 to 5)

$x = 1, \dots N_x$ (total number of latitude bands, 17)

$p = 1, \dots N_p$ (total number of pressure layers, 10 and 6 for the upper and lower stratosphere, respectively)

$t = 1, \dots N_t$ (time in years)

Note that we are not using ALL2 for fingerprint estimation because the ODSonly and GHGonly responses were derived from ALL2. As in previous work (Santer et al. 2003, 2013a), we define the fingerprint $F(x, p)$ by first averaging ozone changes over individual ensemble members, and then calculating the leading empirical orthogonal function (EOF) of the covariance matrix of $\bar{\chi}(x, p, t)$. Many fingerprint studies seek to rotate $F(x, p)$ in a direction that maximizes the signal strength relative to the control run noise (Gillett et al., 2011; Tett et al., 2002). Optimization of $F(x, p)$ generally leads to enhanced detectability of the signal. In this study, we were able to achieve high signal-to-noise levels (see below) without any optimization of $F(x, p)$, and only non-optimized results are discussed. The searched-for fingerprints were computed using ozone anomalies from 1960 to 2016 for ODSonly and GHGonly, and from 1955 to 2014 for ALL1. Sensitivity of our results to the choice of time period for fingerprint calculation was tested by using both a longer period of time (1950 to 2050 for ODSonly and GHGonly) and a shorter time period (1984 to 2016). The fingerprint patterns were found to be relatively insensitive to the choice of time period for estimating $F(x, p)$.

The fingerprint patterns for the upper and lower stratosphere are shown in Figures 5 and 6 (respectively) for ODSonly (A), GHGonly (B), and ALL1 (C). Below each fingerprint is the associated principal component (PC) time series showing the temporal changes in the strength and sign of the pattern in the model simulation. For both the upper and lower stratosphere, the EOF patterns are similar to the latitude-altitude trend patterns presented previously, with GHGonly EOF 1 closely matching the GHGonly linear trends in ozone (Fig. 4A), and the leading ODSonly and ALL1 EOFs closely corresponding to trends computed from the EESC proxy.

In the upper stratosphere, the leading ODSonly and GHGonly EOFs tend to have maximum amplitude at high latitudes in both hemispheres (ODSonly) and in the tropics (GHGonly). These differences are expected based on the spatial structure and time evolution of long-term changes in ozone in ODSonly and GHGonly (see Figs. 4A and B). At the global scale, however, we

note that the EOF 1 patterns in both ODSonly and GHGonly have the same sign at virtually all grid points in the upper stratosphere. The standardized PC time series indicate that the ODSonly and ALL1 results are qualitatively similar to the EESC curve, while PC 1 for GHGonly is more linear. We rely on these PC time series in our subsequent signal-to-noise (S/N) analysis.

In the lower stratosphere, the familiar Antarctic ozone hole is visible in the ODSonly and ALL1 fingerprints (Figs. 6A and C). The GHGonly fingerprint (Fig. 6B) is qualitatively similar to the long-term linear trend in ozone in the GHGonly simulation (Fig. 4A), with ozone changes of opposite sign between 100 to 50 hPa in the tropical lower stratosphere.

A common feature of both the upper and lower stratospheric results in Figs. 5 and 6 is the similarity between the ALL1 and ODSonly fingerprints. This similarity indicates that in the model simulations, changes in ODS are the primary driver of the changes in stratospheric ozone over the past 50 years. A noticeable difference between the upper and lower stratospheric results in Figs. 5 and 6 is that results for the lower stratosphere are noisier. This increased noise is manifest in two ways. First, relative to the upper stratosphere, EOF 1 of ODSonly, GHGonly, and ALL1 consistently explains less of the overall variance in lower

stratospheric ozone changes. Second, the amplitude of the variability of the ODSonly, GHGonly, and ALL1 PC time series is systematically larger in the lower stratosphere. As noted above, these differences between the upper and lower stratosphere reflect the influence of different processes. In the upper stratosphere, local concentrations of ozone are primarily driven by gas-phase chemistry. In contrast, ozone concentrations in the lower stratosphere receive a substantial influence from dynamical transport, which introduces larger interannual variability.

Before presenting the results of the S/N analysis, we first examine the major modes of naturally forced and internal variability estimated from the NAT-h, NAT, and CTL simulations. This is done separately for upper and lower stratospheric ozone (see Figs. 5D-F and 6D-F, respectively). For NAT and CTL we use the full simulations for calculating EOFs and PCs, while for NAT-h we examine the ensemble mean of the individual members prior to EOF and PC estimation, and rely on the years 1960

to 2016. Below each NAT and CTL EOF pattern, we show a 50-year segment of the associated principal component for a 50-year slice, thus facilitating comparisons with the PC time series of ODSonly, GHGonly, and ALL1.

In the upper stratosphere, the 11-year solar cycle is present in NAT (EOF1) and NAT-h (EOF3). EOF1 from the CTL is also manifest in NAT (EOF2), and appears to be associated with interannual changes in tropical upwelling. Tropical upwelling

influences temperature, and therefore also affects ozone via temperature-dependent reaction rate chemistry. In the lower stratosphere, solar-induced ozone changes are not clearly visible in the three leading modes of variability computed from NAT-h, NAT, and CTL. The dominant mode is similar in these simulations, and appears to be associated with changes in upwelling and direct transport of ozone through the lower branch of the BDC. EOF2 of NAT-h displays the 2-3 year QBO influence on lower stratospheric ozone (recall that NAT-h contains an imposed QBO to match observations).

In both the upper and lower stratosphere, the patterns of the dominant modes of variability in NAT-h, NAT, and CTL are noticeably different from the searched-for fingerprint patterns. As noted above, the fingerprints estimated from the ODSonly, GHGonly, and ALL1 simulations show upper stratospheric ozone changes that have the same sign at all grid-points. In contrast, the leading 3 modes of variability estimated from the NAT-h, NAT, and CTL runs do not have the same spatial coherence of ozone change, and are generally characterized by ozone changes of opposite sign at smaller spatial scales. The sole exception is the coherent, same-signed EOF pattern associated the 11-year solar cycle (EOF3 in NAT-h and EOF1 in NAT; see Figs. 5D and E). To quantify these similarities and differences, we calculated pattern correlations between the searched-for fingerprints and the leading noise modes (see Table 1). We rely on these correlations later for interpreting results from the S/N analysis (see below).

## 6 Signal to Noise Estimates

We now seek to identify the model-predicted fingerprints of anthropogenically forced stratospheric ozone changes in observations. We project the time-varying annual-mean latitude-height ozone anomalies from the SWOOSH data, denoted here by $O(x, p, t)$, onto $F(x, p)$ the time-invariant fingerprint from the ODSonly, GHGonly, or ALL1 simulation (see Figs. 5A-C and 6A-C). This projection step yields the signal time series $c\{F, O\}(t)$:

$$c\{F, O\}(t) = \sum_{x=1}^{N_x} \sum_{p=1}^{N_p} F(x, p) O(x, p, t) \,. \tag{1}$$

This projection is equivalent to a spatially uncentered covariance between the patterns $F(x, p)$ and $O(x, p, t)$ at time $t$. The signal time series $c\{F, O\}(t)$ provides information on both the amplitude and sign of the fingerprint in observational data. We can then analyze how $c\{F, O\}(t)$ is changing with time – i.e., whether the searched-for fingerprint pattern is becoming increasingly similar to observed latitude-altitude patterns of ozone change.

Several approaches can be used to assess the significance of the changes in $c\{F, O\}(t)$: direct comparison of actual $c\{F, O\}(t)$ values with a null distribution (e.g. Wigley et al., 1998), or comparison of the trends in $c\{F, O\}(t)$ with a null distribution of trends (e.g. Santer et al., 2003). We use the latter approach here. To assess trend significance, we require a case in which $O(x, p, t)$ is replaced by a record in which we know *a priori* that any spatial correspondence with the fingerprint occurs by chance alone. Here, we use the noise data set $N(x, p, t)$, which is constructed by concatenating together the NAT, NAT-h, and CTL simulations. The associated noise time series $c\{F, N\}(t)$ is the spatially uncentered covariance of $F(x, p)$ and $N(x, p, t)$:

$$c\{F,N\}(t) = \sum_{x=1}^{N_x} \sum_{p=1}^{N_p} F(x,p)N(x,p,t) . \tag{2}$$

Estimates of signal to noise (S/N) are conventionally evaluated (e.g. see Santer et al., 2003) by fitting linear least squares trends of increasing length $L$ to $c\{F,O\}(t)$, and then comparing these trends with the standard deviation of the distribution of

$L$-length trends found in the noise time series $c\{F,N\}(t)$. Signal detection is obtained when the trend in $c\{F,O\}(t)$ exceeds and remains above a stipulated significance level. The test is one-tailed, and assumes a Gaussian distribution of trends in $c\{F,N\}(t)$. Here we use the 1% significance level, which corresponds to a S/N ratio close to 2.33. S/N ratios that consistently exceed this level are highly unlikely to be due to the combined effects of natural internal variability and natural external forcing.

Since the sign of the fingerprint EOFs in Figs. 5 and 6 is arbitrary (and since the EOF patterns are very similar to the patterns of ozone trends in Fig. 4), we stipulate that the sign of the leading EOF in ODSonly, GHGonly, or ALL1 should match the sign of the corresponding simulation's ozone trend pattern. With this stipulation, negative S/N ratios indicate that the sign of the overall change in ozone in the anthropogenically forced simulation is inconsistent with the observed ozone change.

We estimate S/N ratios by fitting least-squares trends to $L$-length segments of $c\{F,O\}(t)$, and then comparing these signal trends with $s(L)$, the standard deviation of the sampling distribution of $m$ maximally overlapping $L$-length trends in $c\{F,N\}(t)$ (i.e., for overlap by all but one year). As noted earlier, $c\{F,N\}(t)$ is the concatenation of NAT, NAT-h, and CTL, and due to the discontinuities between the simulations, we discard trends that span two different noise simulations. As $L$ increases, $m$ decreases. We use maximally overlapping trends to guard against excluding the largest changes in ozone from our analysis.

We use a minimum trend length of for $L = 10$ years, so the first S/N ratio (and the earliest possible detection time) is for 10-year trends starting in 1984 and ending in 1993.

One innovative feature of this work is that we derive estimates of S/N behavior using two types of approach. The regression is of the standard form:

$$y = X\beta + \epsilon , \tag{3}$$

where $y$ can be either $c\{F,O\}(t)$ or $c\{F,N\}(t)$, $X$ contains the regressors, $\beta$ contains the reported trend/regression coefficients, and $\varepsilon$ is an error term.

The first method we employ for S/N estimation is of the conventional form used in such studies (see, e.g., Santer et al., 2003), where $X$ is the time coordinate in years:

$$X = \begin{pmatrix} 1 & t_0 \\ \vdots & \vdots \\ 1 & t_0 + L \end{pmatrix}, \quad \beta = \begin{pmatrix} \beta_0 \\ \beta_l \end{pmatrix}, \tag{4}$$

$t_0$ is the starting year (1984), $\beta_0$ is the intercept, and $\beta_l(L)$ is the reported trend coefficient of interest, with units of percent
change in ozone per year.

The second method relies on linear regression between $c\{F,O\}(t)$ or $c\{F,N\}(t)$ and a selected $PC_1$ time series. There are a total of six $PC_1$ time series, one for each layer (upper and lower stratosphere) and for each of the ODSonly, GHGonly and ALL1 fingerprints. In this second method, the regressor $X$, contains the $PC_1$ time series for the selected trend years:

$$X = \begin{pmatrix} 1 & PC_1(t_0) \\ \vdots & \vdots \\ 1 & PC_1(t_0 + L) \end{pmatrix}, \quad \beta = \begin{pmatrix} \beta_0 \\ \beta_{PC_1} \end{pmatrix}, \tag{5}$$

where $\beta_{PC_1}(L)$ is the reported trend coefficient, and has units in percent change per unit standard deviation of $PC_1$. For example, if both $c\{F,O\}(t)$ and $PC_1$ show similar nonlinear behavior over a common analysis period (such as 1984 to 2016),
the regression coefficient $\beta_{PC_1}(L)$ will be unusually large relative to values of the regression coefficient estimated with $c\{F,N\}(t)$. In cases where $PC_1$ exhibits change that is nearly linear with time, then the change in $\beta_l(L)$ (the linear trend representation of ozone change) and $\beta_{PC_1}(L)$ with increasing $L$ will be similar.

The advantage of the second method is that it accommodates forcings whose time evolution is markedly nonlinear over the
observational record, whereas the first method implicitly assumes that the forcing evolution is quasi-linear over the period of interest. Like the first method, however, the second method still assumes that the spatial structure of the searched-for fingerprint is essentially unchanged with time (an assumption that is justifiable; see above). In summary, our second method of estimating S/N behavior uses information on both the model-derived fingerprint and its time evolution to search for the fingerprint in SWOOSH data. Since the ODSonly and GHGonly fingerprints have different time evolution properties (particularly in the
upper stratosphere; compare Figs. 5A and B), explicit consideration of this time evolution information can be useful in separating the ODS- and GHG-induced ozone change signals in observations. In the following, we refer to our first S/N method as the *linear* or $\beta_l$ trends method, and refer to the second method as either the $PC_1$ or nonlinear signal method.

The S/N results are shown in Figures 7 and 8 for the upper and lower stratosphere respectively. The left panels show S/N
estimates from the $\beta_l$ trends, and the right panels show the S/N estimates based on $\beta_{PC_1}$ trends. Results are for the three fingerprints: ALL1, GHGonly and ODSonly. Since the $\beta_{PC_1}$ trends are regressions to the associated $PC_1$ of the fingerprints,

the ALL1 simulation (which ends in December 2014) yields S/N ratios that end 2014, while the ODSonly and GHGonly simulations span the full observational record, allowing S/N ratios to be calculated through to 2016 (inclusive).

We consider first S/N ratios for the upper stratosphere, and for results based on the first method of estimating trends (see left
panels of Fig. 7). The use of $\beta_l$ yields virtually identical signal trends, noise trends, and S/N ratios for the ALL1 and ODSonly fingerprints. Recall that the signal trends are calculated from the time series of the projections of the SWOOSH data onto the fingerprints. The ALL1 and ODSonly signal trends are large and positive in the first 10-20 years. These large positive trends arise during the depletion era, when upper stratospheric ozone decreased steadily until the late 1990s. The positive sign of the signal trends that end in the depletion era reflects the consistency in sign between the observed ozone loss and the loss of ozone
captured in the ALL1 and ODSonly simulations (see Figs. 5A and C). As the trend length $L$ extends into the 21$^{st}$ century, the signal trends for ALL1 and ODSonly decline towards zero. This is a result of the stabilization of ozone loss and the emerging recovery of stratospheric ozone. Such nonlinear behavior is not well-described by fitting a straight line through the entire $c\{F, O\}(t)$ time series.

The time evolution of the $\beta_l$ signal trends obtained with the GHGonly fingerprint mirrors the results for ODSonly and ALL1, but is of opposite sign (Fig. 7A). The negative sign of the GHG signal trend arises because of the sign mismatch between upper stratospheric ozone changes in SWOOSH and the GHGonly simulation (see Fig. 1A). The standard deviations of the noise trends (Fig. 7B) are very similar for the 3 fingerprints, which is expected given the similarity between the ODSonly, GHGonly, and ALL1 fingerprint patterns (see Table 1). The decrease in the amplitude of the noise trends with increasing trend-fitting
period is typical behavior for many different climate variables (see, e.g., Santer et al. 2013a,b).

Because of early 21$^{st}$ century ozone recovery, and the impact of recovery on $\beta_l$ signal trends, S/N ratios obtained with the ODSonly and ALL1 fingerprints decline from the late 1990s through 2016 (2014 in the case of ALL1; see Fig. 7C). For both of these fingerprints, however, S/N ratios remain above the stipulated 1% significance threshold, even for signal trends
sampling the recovery era. This result suggests that even with an emerging "healing" signal in the early 21$^{st}$ century, the overall loss in upper stratospheric ozone over the entire 1984 to 2014 analysis period is still significantly larger than can be explained by WACCM-based estimates of internally generated and solar forced changes in ozone. The close agreement between the ALL1 and ODSonly S/N ratios indicates that human-caused changes in stratospheric ozone are the dominant contributor to forced ozone changes in ALL1. In the GHGonly case, the S/N ratio becomes less negative as the trend length $L$ increases. This
reflects the observed increase in upper stratospheric ozone in the early 2000s, which projects positively onto the GHGonly fingerprint (Fig. 5B). In summary, use of a standard fingerprint identification method allows us to positively detect (in SWOOSH ozone data) the upper stratospheric ozone fingerprints in response ODS forcing alone and in response to combined ODS and GHG forcing. For all three fingerprints, however, the $\beta_l$ signal trends display pronounced nonlinear behavior because

of the observed "depletion followed by recovery". This nonlinear behavior is not accounted for in the first ($\beta_l$ based) signal detection method, and decreases S/N ratios as the analysis period lengthens.

To address this problem, we also calculated S/N ratios using the above-described method 2, which relies on regressing the $c\{F, O\}(t)$ time series onto each of the principal component time series for the individual ODSonly, GHGonly, and ALL1 fingerprints (see Figs. 5A-C). These "signal" regression coefficients, $\beta_{PC_1}$, are shown as a function of $L$, the analysis period length (Fig. 7; D-F). It is instructive to compare the S/N results on the left and right panels of Fig. 7, which highlights differences between the linear and nonlinear signal detection approaches (i.e., between methods 1 and 2, respectively). As $L$ increases and the analysis periods sample both ozone depletion and recovery, use of method 2 markedly increases S/N ratios relative to a purely linear representation of signal trends. This enhancement of S/N occurs because method 2 incorporates information about the nonlinear behavior of the upper stratospheric ozone signal (behavior that is common to the real world and the WACCM ODSonly and ALL1 simulations). Further information on the enhancement of S/N is provided in Table 2.

There are several other noteworthy features of the method 2 results in Fig. 7. First, the method 1 and method 2 S/N ratios for the GHGonly fingerprint show qualitatively and quantitatively similar behavior. This similarity is due to the fact that $PC_1$ for the GHGonly fingerprint (Fig. 5B) is well-described by a linear trend, so the regression coefficients between the linear $PC_1$ and the nonlinear $c\{F, O\}(t)$ time series change markedly as $L$ increases. Second, $S_L$, the standard deviation of the sampling distribution of noise trends, shows some differences in the method 1 and method 2 cases (Fig. 7B and E). In method 1, $S_L$ decreases in amplitude with increasing $L$; this holds for all three fingerprints. In contrast, the method 2 results show that for the ODSonly and ALL1 fingerprints, $S_L$ has a local minimum for $L$=18 years, and then increases slightly for longer analysis periods. This local minimum in $S_L$ is absent in the GHGonly results for method 2.

The differences between the method 1 and method 2 $S_L$ results are probably related to multiple factors. These include: 1) the strong influence of solar forcing on upper stratospheric ozone; 2) the periodicity in solar forcing in the NAT simulation; 3) the fact that the dominant mode of ozone variability in NAT is more similar to the ODSonly and ALL1 fingerprint patterns than to the GHGonly fingerprint (see pattern correlation results in upper triangle of Table 1); and 4) the quasi-linearity of the $PC_1$ time series for the GHGonly fingerprint, and the nonlinearity of the $PC_1$ time series for the ODSonly and ALL1 fingerprints (see Fig. 5).[1]

---

[1] Recall that for method 2, the "noise" regression coefficients are calculated between maximally overlapping $L$-year segments of $c\{F, N\}(t)$ (the projection of the combined NAT and CTL data onto the ODSonly, GHGonly, or ALL1 fingerprint) and an $L$-year segment of the $PC_1$ time series for the ODSonly, GHGonly, or ALL1 fingerprint. As the $L$-year analysis window is being advanced through $c\{F, N\}(t)$, there will be (for certain values of $L$) times of random agreement in the phasing of solar variability and the nonlinear behavior in the ODSonly and ALL1 $PC_1$ time series. Such agreement would be expected to yield more complex behavior in $S_L$ as a function of $L$.

For the upper stratosphere, we also explored whether our S/N results were robust to the choice of ozone dataset. This involved replicating the SWOOSH-based S/N analysis with the SBUV_CDR upper stratospheric ozone dataset, which was available from 1979 to 2015. The SBUV S/N analysis was performed with the same model fingerprints shown in Figs. 5A-C. Relative to the SWOOSH results, S/N ratios were generally slightly larger when calculated with SBUV data. As in the SWOOSH case, the SBUV results show that use of method 2 yields a noticeable enhancement of S/N ratios for the ODSonly and ALL1 fingerprints (see Figure S5 in supplementary material). In summary, the method 2 results for the upper stratosphere show (relative to method 1) higher S/N ratios and more confident identification of the ODSonly and ALL1 fingerprints in observations. Even with method 2, however, the GHGonly ozone signal cannot be detected by the final year of the SWOOSH record (2016).

Next, we examine the lower stratosphere. Because of the above-described differences in the post-2005 behavior of tropical lower stratospheric ozone in ALL1 and SWOOSH (see Fig. 1C), we partition our lower stratospheric S/N analysis into two cases: for a global domain, and for a domain poleward of 30°S to 30°N (i.e., excluding the tropics). Figures 8A-F show the signal trends, noise trends, and S/N ratios for the lower stratosphere, estimated with methods 1 and 2. Consider the $\beta_l$ signal trends first. As in the case of the upper stratosphere, the behavior of $\beta_l$ as a function of the trend length $L$ is similar for $c\{F, O\}(t)$ time series obtained with the ODSonly and ALL1 fingerprints. These similarities are greater when the tropics are excluded from the analysis. Inclusion or exclusion of the tropics also impacts the $\beta_l$ signal trends obtained with the GHGonly fingerprint: in the global analysis, trends ending after 2000 are slightly positive, while in the "tropics excluded" case, trends ending after 2000 are slightly negative. The S/N ratios for the $\beta_l$ trends (Fig. 8 – bottom left panel), show that in the lower stratosphere, both the ALL1 and ODSonly fingerprint patterns are detectable in the SWOOSH observations. This holds for the global and "tropics excluded" domains. The GHGonly fingerprint is not statistically identifiable in either domain by the end of the SWOOSH record.

Recall that in the upper stratosphere, the use of method 2 yielded S/N ratios for the ODSonly and ALL1 fingerprints that were markedly larger than for method 1. A large enhancement of S/N ratios is not evident in the lower stratosphere (Fig. 8 C&F). A small S/N enhancement occurs for ODSonly and ALL1 fingerprints, but only when the tropics are excluded. The smaller differences between the method 1 and method 2 S/N results (relative to the upper stratosphere) are likely due to the larger variability of ozone in the lower stratosphere (see Fig. 3). As in the case of method 1, use of method 2 leads to consistent detection of the ODSonly and ALL1 fingerprints in observed lower stratospheric ozone data, but does not yield positive identification of the GHGonly fingerprint.

In summary, we find that for both the upper and lower stratosphere, the observed decline and emerging recovery of stratospheric ozone is strongly influenced by secular changes in anthropogenic chlorofluorocarbons. These secular changes reflect scientific recognition of the serious consequences of ozone depletion, and the eventual formulation and implementation

of the Montreal Protocol. We were able to identify the model-predicted latitude/altitude patterns of ozone depletion in observations of upper and lower stratospheric ozone loss. We were not able to identify the model-predicted ozone changes arising from human-caused increases in well-mixed GHGs. Our results show that in the upper stratosphere, ozone depletion and recovery introduce pronounced nonlinearity in both modeled and observed ozone changes. Accounting for this common nonlinear behavior in our signal detection method (in our method 2) substantially amplifies S/N ratios. This amplification is largest in the upper stratosphere, where the nonlinear behavior is clearest and noise levels are lowest (see Table 2).

## 7 Discussion

In this study, we used the SWOOSH stratospheric ozone dataset and simulations performed with the Whole Atmosphere Community Climate Model (WACCM) to evaluate the relative detectability of ozone changes arising from forcing by ozone depleting substances (ODS), greenhouse gases (GHG), and combined changes in ODS and GHG. Our focus was on the period covered by the SWOOSH data (1984 to 2016). Our detection study relied on zonal-mean latitude-height profiles ("fingerprints") of ozone change. The lower and upper stratosphere were considered separately in this investigation. This separation was made because the forcing by ODS and GHG has different ozone-change signatures in the lower and upper stratosphere, and because the amplitude and patterns of ozone variability differ in these two atmospheric regions.

The credibility of our findings rests on the reliability of the WACCM-based estimates of natural climate variability. Before conducting our signal detection study, we first investigated the skill of the WACCM model in capturing the observed variability of global-mean lower and upper stratospheric ozone. This comparison was performed using high-pass and band-pass filtered data, which isolated variability on short (less than 3 years) and long (5-20 years) timescales, respectively. We found that in the lower stratosphere, the WACCM simulation incorporating variability associated with the QBO yielded short-term variability that was closest to the SWOOSH-based estimate. Reliable simulation of the longer-timescale variability of ozone required the inclusion of solar forcing, particularly in the upper stratosphere (where solar-forced ozone changes are largest). In the simulation of historical climate change with combined ODS and GHG forcing (which is the simulation most relevant for comparison with observations), the amplitude of the longer-timescale variability is larger than in SWOOSH in the lower stratosphere, and slightly smaller than in SWOOSH in the upper stratosphere.

In global-mean terms, the upper and lower stratospheric ozone changes from the ODS+GHG simulation best matches the observed 20[th] century ozone depletion and early 21[st] century ozone recovery (see Fig. 1). In the full pattern analysis of zonal-mean latitude-altitude changes in ozone, all three anthropogenic fingerprints (ODS, GHG, and ODS+GHG) have coherent, large-scale structure. The three fingerprints are spatially dissimilar to the smaller-scale (and opposite-signed) structure of the dominant modes of internal and solar variability. The ODS and GHG fingerprints have similar large-scale structure in the

upper stratosphere, but are of opposite sign, and have distinctly different time evolution over the observational record (see Figs. 5A-C).

We applied two different fingerprint identification methods. The first (our method 1) has been routinely used in a number of detection and attribution studies. It assumes that the time evolution of the fingerprint pattern is quasi-linear over the length of the observational record. In the case of stratospheric ozone, however, the time evolution of the ODS and ODS+GHG fingerprints (which we refer to as "signal time series") is markedly nonlinear because of 20[th] century stratospheric ozone depletion and early 21[st] century ozone recovery. This means that method 1 – which relies on linear fits to both signal and noise time series[2] – is sub-optimal for assessing the S/N properties of stratospheric ozone changes.

In contrast, our method 2 explicitly accounts for the nonlinear time evolution of the ODS and ODS+GHG fingerprints. In the upper stratosphere, where this nonlinear behavior is clearest, method 2 yields S/N ratios for ODS and ODS+GHG fingerprints that are markedly larger than those obtained with method 1. In both the upper and lower stratosphere, only the ODS+GHG and ODS only fingerprint patterns were detectable before the current final year (2016) of the SWOOSH ozone dataset. The GHG-driven fingerprint of ozone changes was not statistically identifiable in the either the upper or lower stratospheric SWOOSH data, irrespective of whether we employed method 1 or method 2.

Our results illustrate the importance of explicitly considering the forced, nonlinear temporal changes in the signal of interest. Here, both the 20[th] century ozone depletion and the emerging early 21[st] century ozone recovery are driven by well-understood temporal changes in ODS forcing. The time structure of the ODS ozone signal is very different from the much more linear increase in ozone caused by increases in GHG levels (see Figs. 5A-C). Our findings suggest that these differences in time evolution are key to confident separation of the ODS and GHG ozone signals. While one previous detection and attribution study involving stratospheric ozone considered time information, it was not able to clearly separate ODS and GHG signals (Gillett et al., 2011). The key advantage here is that we are dealing with an observational record that extends to 2016 compared to 2005 in Gillett et al., and thus are better to capture the emerging recovery of stratospheric ozone, and the nonlinear behavior of the ODS-driven ozone signal. Figure 9 shows a schematic comparing method 1 and our method 2 which can be applied to attribution studies where there is evidence that the climate forcing has a temporal component that is changing nonlinearly over historical records. Although our study was confined to using spatial fingerprints in the latitude/altitude domain, the method is equally suitable for other domains, such as for longitude/latitude fingerprints. Not shown in Fig. 9 is the calculation of the standard deviation of the sampling distribution for the "noise" regression coefficients, $s(L)$. Noise calculations use the same approach applied for the signal: projection of the combined noise simulation data onto the fingerprint to retrieve c{F,N}(t),

---

[2]As described above, "noise time series" are obtained by projecting model estimates of naturally caused changes in ozone (both internally generated, and arising from solar forcing) onto the ODS, GHG, and ODS+GHG fingerprints.

followed by  the calculation of regression coefficients. The regressions rely on *L*-year segments of either c{F,N}(t) and time (in the case of method 1) or c{F,N}(t) and the leading PC time series of the model ALL1, ODSonly, or GHGonly simulation (in the case of method 2).

As noted in section 2, there is a discrepancy between simulated and observed SWOOSH tropical lower stratospheric ozone post-2005. This is also a region where there is considerable divergence between different observational estimates of ozone changes. (WMO,2014). Both of these factors (model-data differences and observational uncertainty) motivated the additional analysis with exclusion of ozone changes in the tropical lower stratosphere from our S/N analysis. In the "tropics excluded" case, there is high confidence in our detection of the model ODS signal in the lower stratosphere. For the two reasons outlined

above, we have less confidence in the interpretation of our S/N results for the global "tropics included" case, especially for method 2 where the PC time series are used. Due to the noticeable divergence between simulated and observed post-2005 ozone changes in the tropical lower stratosphere, inclusion of the tropics reduces S/N ratios for our method 2 (i.e., the temporal evolution of ozone differs over for the last decade in observations and in the ODSonly signal, thus reducing the regression coefficient in method 2).

The above-described results pertain to the identification of the model anthropogenic fingerprints in observations. A related question is the detection time of the GHG signal – the point in the future at which the currently undetectable GHG signal might become identifiable if we continued to monitor stratospheric ozone. We can obtain a purely model-based estimate of this detection time by using the ALL2 ensemble members that extend to the end of the 21$^{st}$ century, and treating these ensemble

members as surrogate observations.

As noted above, S/N ratios for the GHG fingerprint were approaching the 1% significance level (our stipulated signal detection threshold) near the end of SWOOSH observational record (see Figs. 7 and 8). To investigate the expected GHG fingerprint detection time in "model world", we use the same upper and lower stratospheric GHG fingerprint patterns that we searched

for in the SWOOSH ozone data (see Figs. 5B and 6B). We also estimate the "model world" detection time for the two ODS fingerprints (Figs. 5A and 6A). In each case, ozone data from individual realizations of the ALL2 simulation are projected onto the GHG and ODS fingerprints yielding the signal time series for our method 1 S/N analysis. For the method 2 S/N analysis, we additionally projected the 20$^{th}$ and 21$^{st}$ century GHG (ODS) ozone anomaly data onto the GHG (ODS) fingerprints, and then regressed the projection time series obtained with ALL2 onto the projection time series obtained with

the GHG (ODS) data. The method 1 and method 2 estimates of the denominator of the S/N ratio were calculated as described in Section 6, but are now computed for analysis periods spanning the range 10 to 67 years.

Results in Figure 10 are combined for the upper and lower stratosphere. The "model only" analysis clearly illustrates the critical importance of accounting for nonlinearity in the behavior of the ODS signal. Linear trends provide a reasonable fit to

the initial ozone depletion phase. As *L* increases, however, linear trends are not an accurate representation of the nonlinear "ozone depletion followed by ozone recovery". In method 1, therefore (which relies on linear trends), the "model world" S/N ratio for the ODS fingerprint is initially above the detectability threshold, but then dips and remains below this threshold. This transition from a significant to a non-significant result occurs around 2012 for the upper stratosphere and between 2010 and

5    2020 in the lower stratosphere (see Figs. 10A and C). Since the time evolution of the GHG-induced stratospheric ozone changes is more linear, method 1 yields "model world" detection of the GHG fingerprint by roughly 2025 in the upper stratosphere and by 2012-2014 in the lower stratosphere.

Because method 2 accounts for nonlinearity in the time evolution of the ODS signal, the "model world" S/N ratios remain

10    above the stipulated 1% significance threshold for all values of the analysis period *L*, both for the upper and the lower stratosphere (see Figs. 10B and D). This is a strikingly different result from the method 1 S/N ratios for the ODS fingerprint. The GHG fingerprint is also consistently identifiable in method 2, with "model world" detection times similar to those estimated for method 1. The model results suggest, therefore, that the GHG fingerprint may not be identifiable in the SWOOSH data for at least another 5 to 10 years.

Our study relies on a single global climate model (WACCM). It would be important to determine whether the S/N results obtained here are consistent with those inferred from other climate models. Such a multi-model assessment will require use of models with a well-resolved stratosphere, and with reasonable representation of observed ozone variability. This will be the focus of subsequent work.

**Acknowledgements**

The authors would like to thank Sean Davis and Karen Rosenlof, from the National Oceanic & Atmospheric Administration, Chemical Sciences Division, for access to the SWOOSH ozone dataset. The National Center for Atmospheric Research is funded by the National Science Foundation.

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

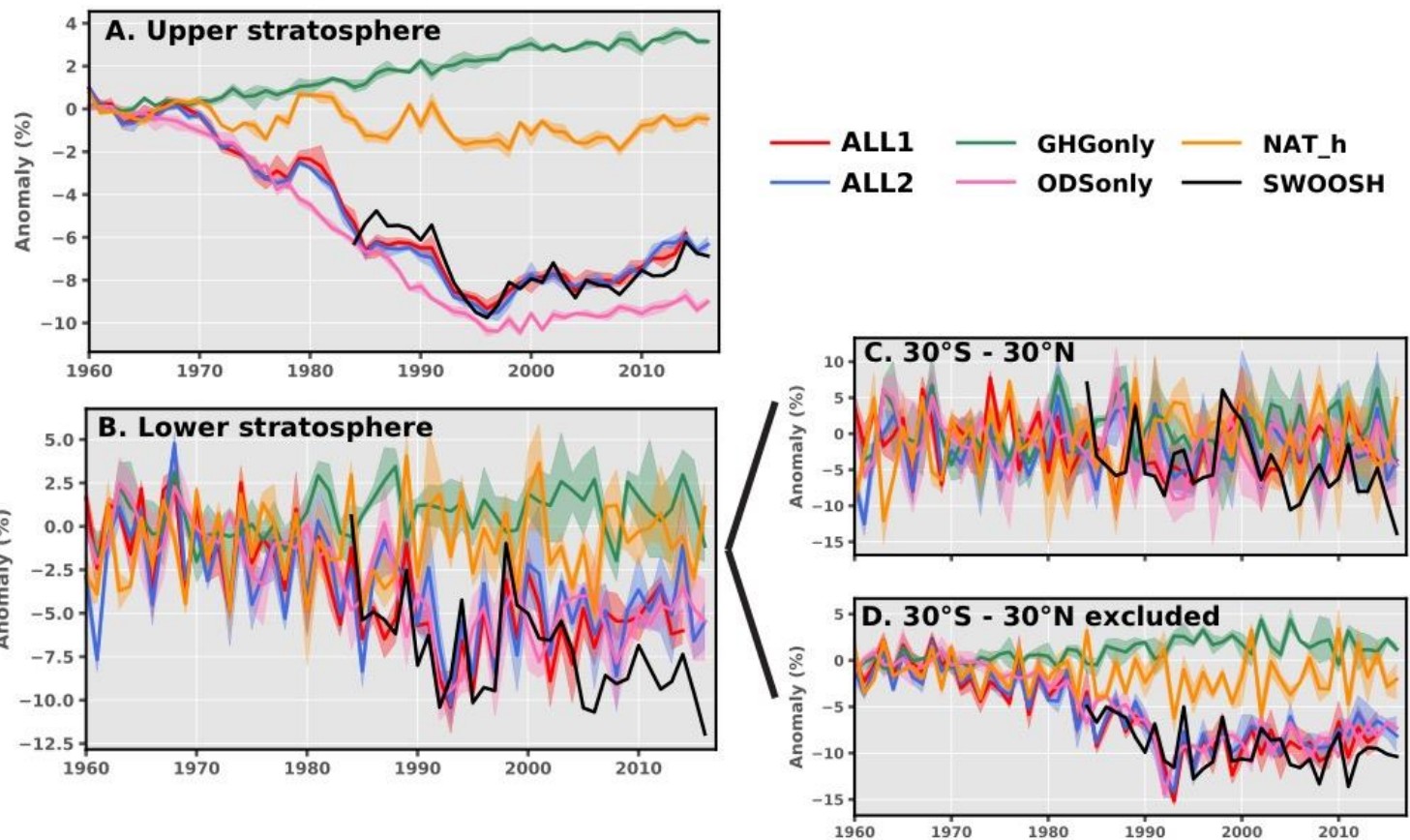

Figure 1: Time series of global mean annual ozone anomalies for A) the upper stratosphere (1-10 hPa) and B) the lower stratosphere, (40-100 hPa). The right panel shows the lower stratosphere for C) the tropics (30ºS to 30ºN) and D) tropics excluded. Annual anomalies were calculated from monthly-mean model output and SWOOSH ozone dataset, and both were weighted by latitude and pressure. All model anomalies are shown as percent differences from the base years of 1960-1969 in ALL1. For model simulations, the spread shows the upper and lower bounds of all ensemble members. The SWOOSH record spans the period from 1984-2016 and the anomalies were also defined as percent differences from the base years of 1960-1969 in the ALL1 simulation.

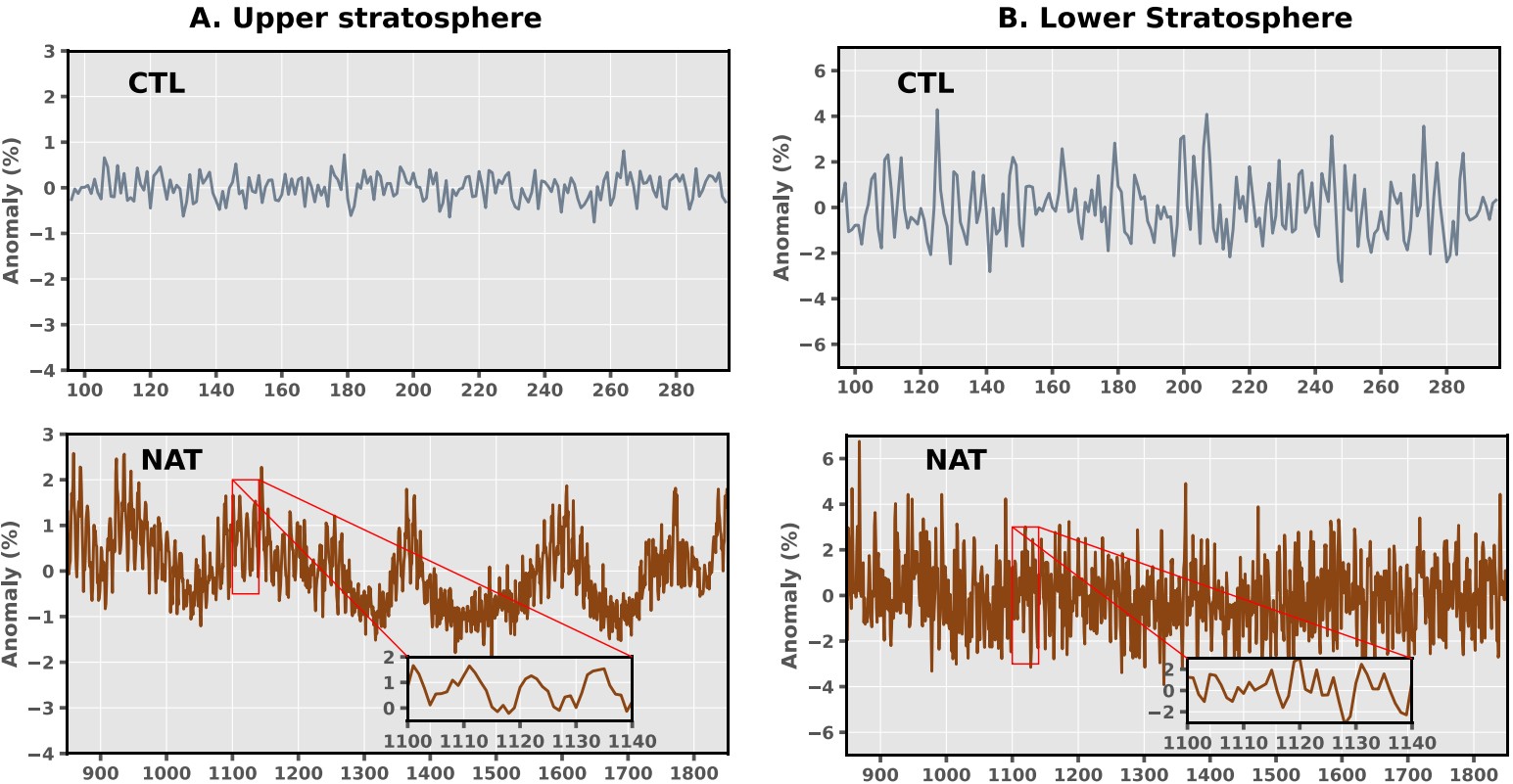

**Figure 2: Same as Fig. 1 but for ozone anomalies from the pre-industrial control simulation (top), and the pre-industrial natural simulation (bottom) for the upper (A) and lower stratosphere (B). The NAT results show (as an inset) an expanded 40-year segment of the anomaly data.**

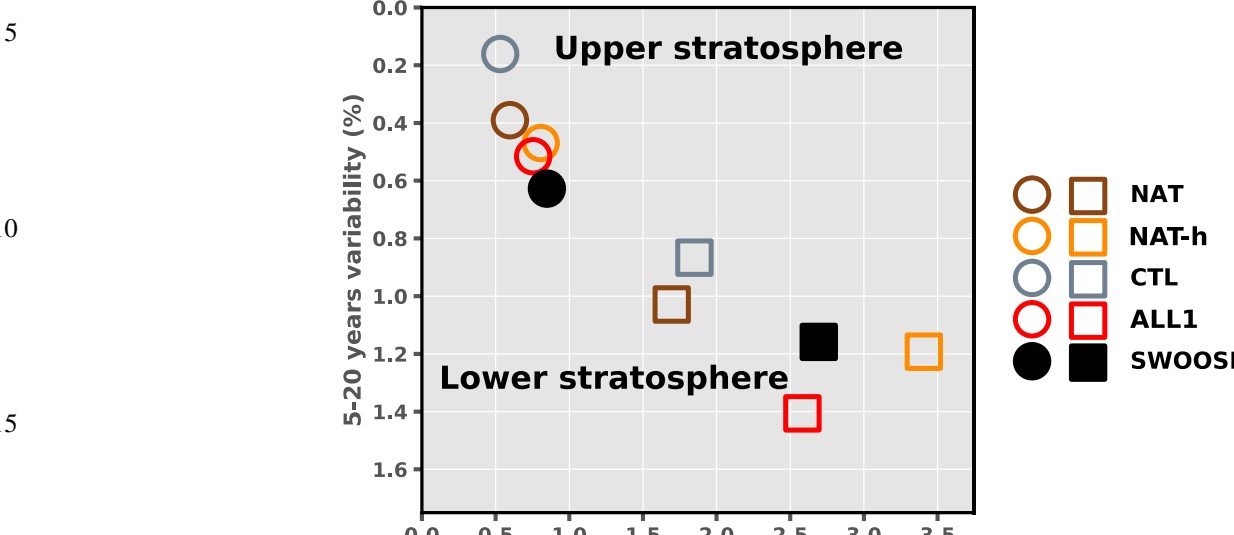

**Figure 3: Comparison of simulated and observed temporal variability of global ozone anomalies in the upper and lower stratosphere. Results for the lower (upper) stratosphere are represented by square markers (circles). Variability on monthly to interannual timescales is plotted versus variability on timescales of 5-20 years. After removal of long-term trends from raw anomalies with a 30 year low-pass filter, a Butterworth filter was used to perform band-pass and high-pass filtering. There is no overlap between the frequencies isolated by the high- and band-pass filters. Results for NAT, CTL and NAT-h are from 1000, 200 and 168 years respectively, while ALL1 and SWOOSH are from 1984-01 to 2014-12 and 2016-12 respectively. Temporal standard deviations for the band- and high-pass filtered data are given as the percent differences from the base years of 1960-1969 in ALL1. .**

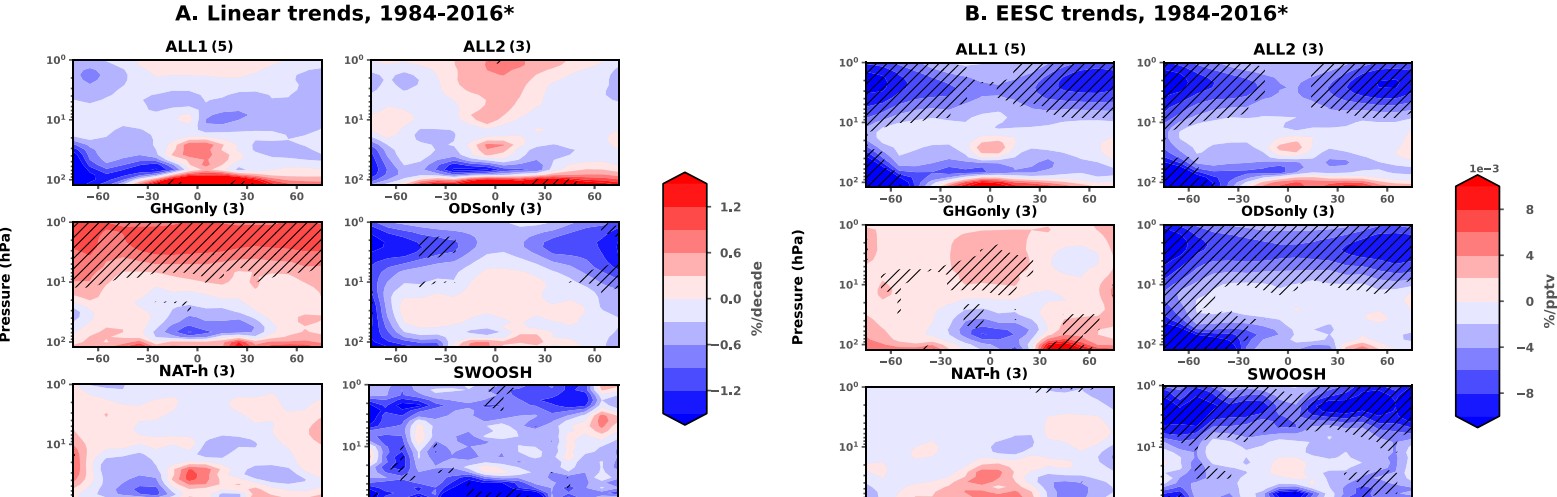

**Figure 4: Zonal-mean stratospheric ozone trends estimated using A) linear regression and B) EESC regression of annual-mean anomalies from 1984-2016. For model simulations, trend estimates use the ensemble mean. Hatching indicates significance at the 5% level or better using a two-sided Student's t-test. Figures in brackets indicate the number of ensemble members. All of the trends shown are for 1984-2016, except for ALL1 which is from 1984-2014.**

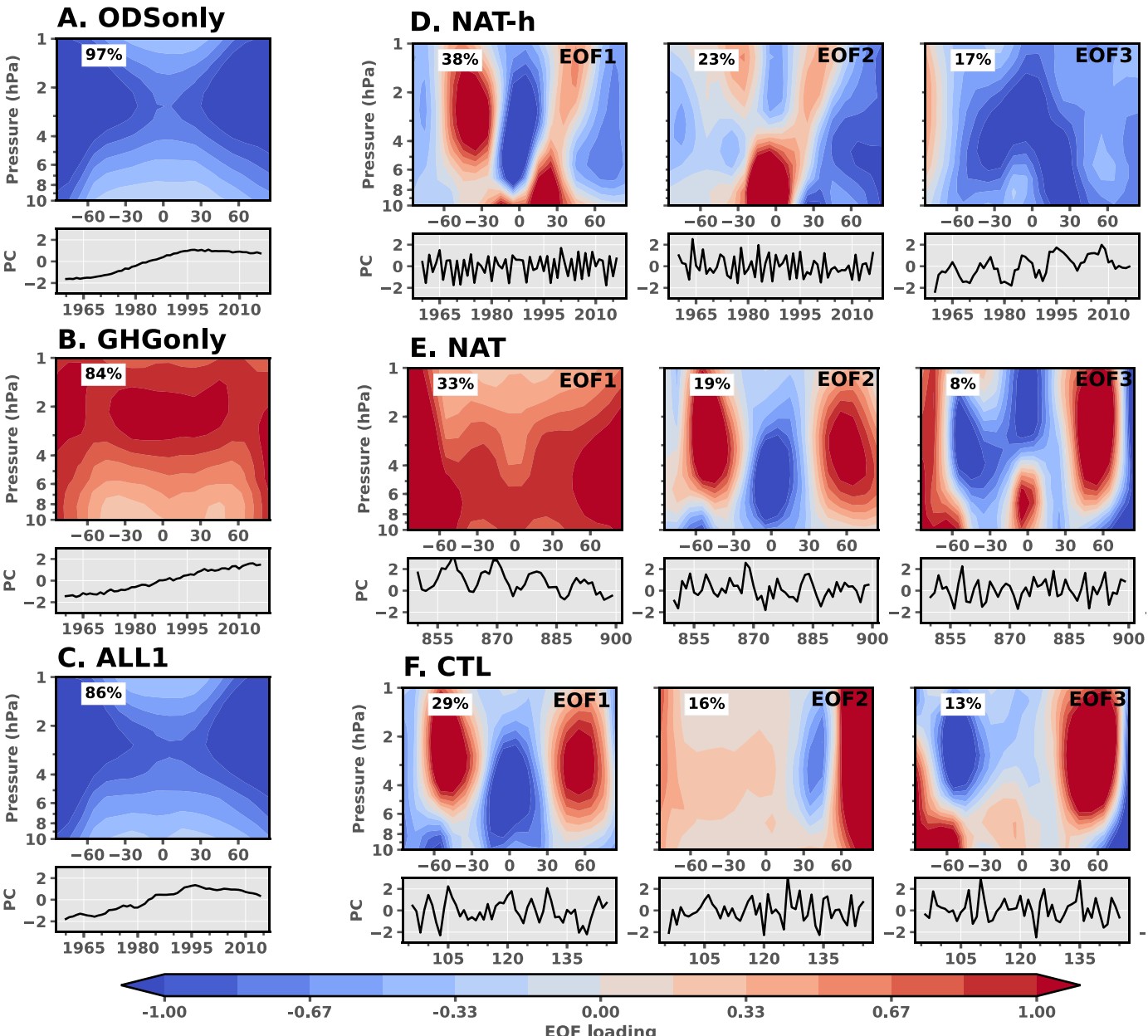

**Figure 5: Leading signal and natural variability modes for upper stratospheric ozone in WACCM. The signal modes are the leading EOFs of the ensemble mean anomalies from GHGonly, ODSonly and ALL1 calculated over 1960-2016 for GHGonly, and ODSonly and over 1960-2014 for ALL1. The leading natural variability modes are EOFs 1, 2 and 3 of the NAT, NAT-h, and CTL. The percentage variance explained by each mode is given in the upper left corner of the EOF. For NAT-h the EOFs where calculated from the ensemble-mean anomalies for the period 1960-2016. The total lengths of the NAT and CTL simulations are 1000, and 200 years respectively. Recall that NAT-h is defined by subtraction (see Section 2).**

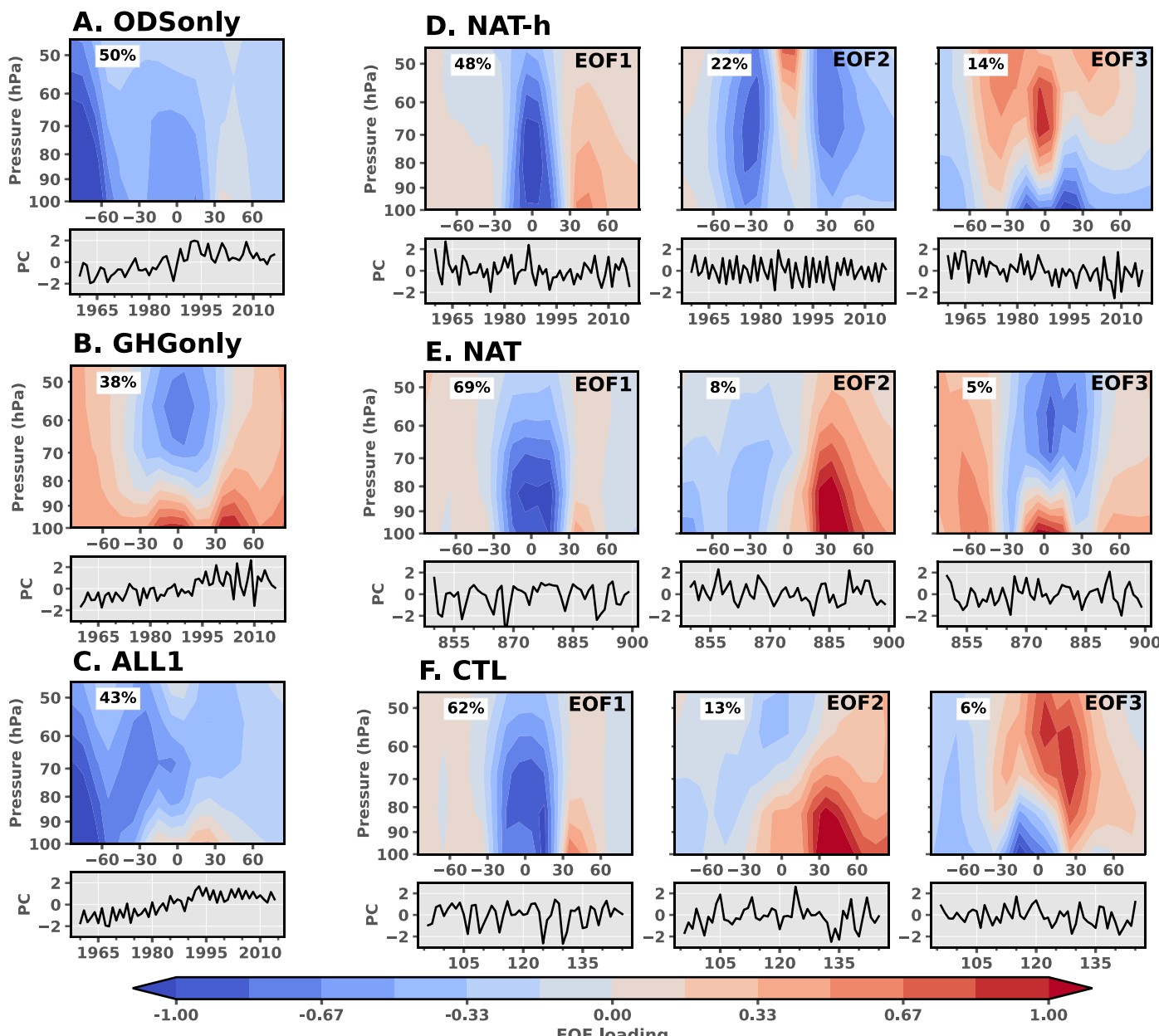

**Figure 6: Same as Fig. 5 but for the lower stratosphere.**

# Upper Stratosphere

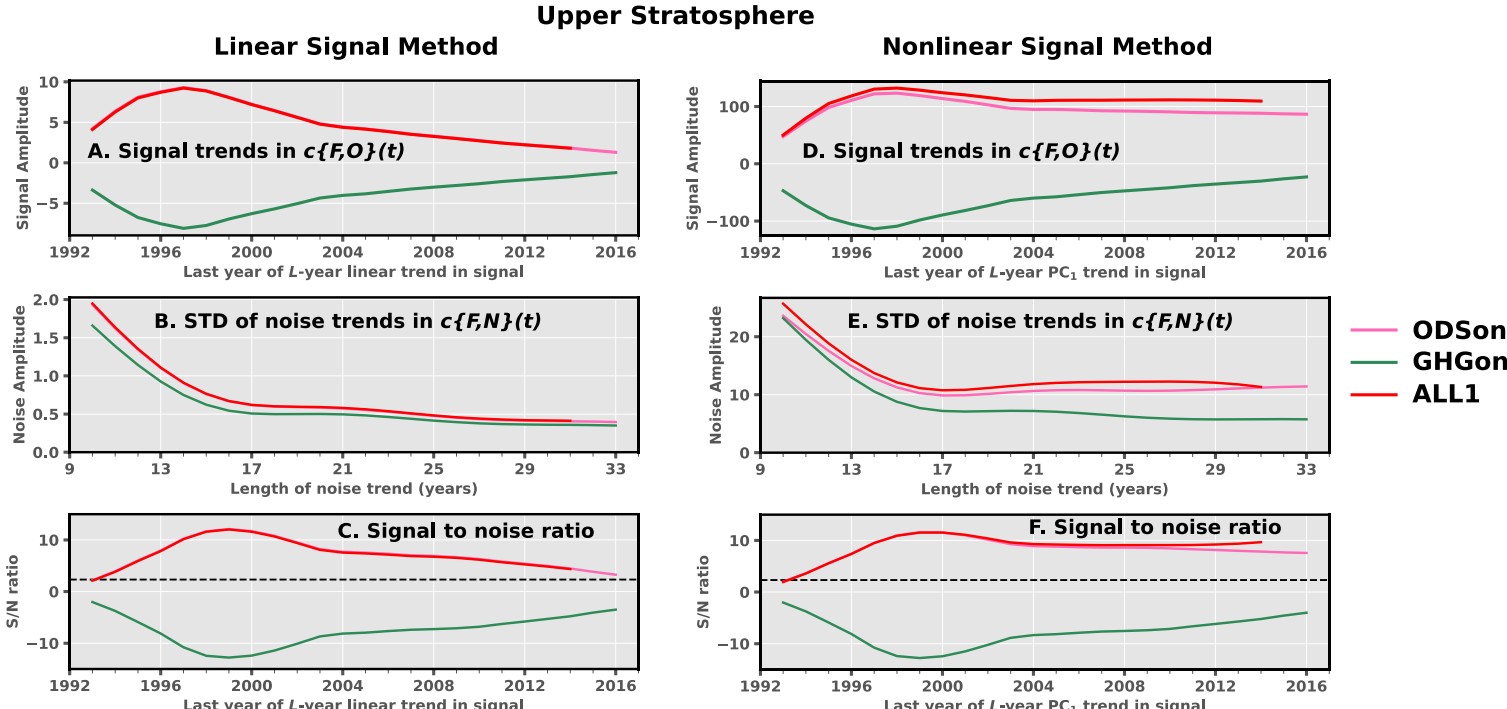

**Figure 7: Results from the S/N analysis of simulated and observed changes in zonal-mean upper and lower stratospheric ozone.** Signal time-series provide information on the similarity between the time-invariant ODSonly/GHGonly/ALL1 fingerprint pattern and the time-varying observed patterns of stratospheric ozone change. Signal detection relies on both simple linear regression (A-C; "method 1") and regression between $c\{F,O\}(t)$ and the leading PC of the ODSonly, GHGonly, or ALL1 simulation (D-F; "method 2"). Results are a function of the analysis period $L$ (in years). The $L$-year trends in the method 1 and method 2 signal time series are plotted in the top row. The year on the abscissa is the end year of the $L$-length trend beginning in 1984. Noise time series indicate the level of similarity between the searched-for ODSonly, GHGonly, and ALL1 fingerprints and the concatenated CTL, NAT-h, and NAT estimates of natural variability of ozone. The middle panels show the standard deviation of the distribution of maximally overlapping $L$-year trends in $c\{F,N\}(t)$. The S/N ratio is shown in the bottom panels. The dashed horizontal line indicates the stipulated significance level of 1% for signal detection.

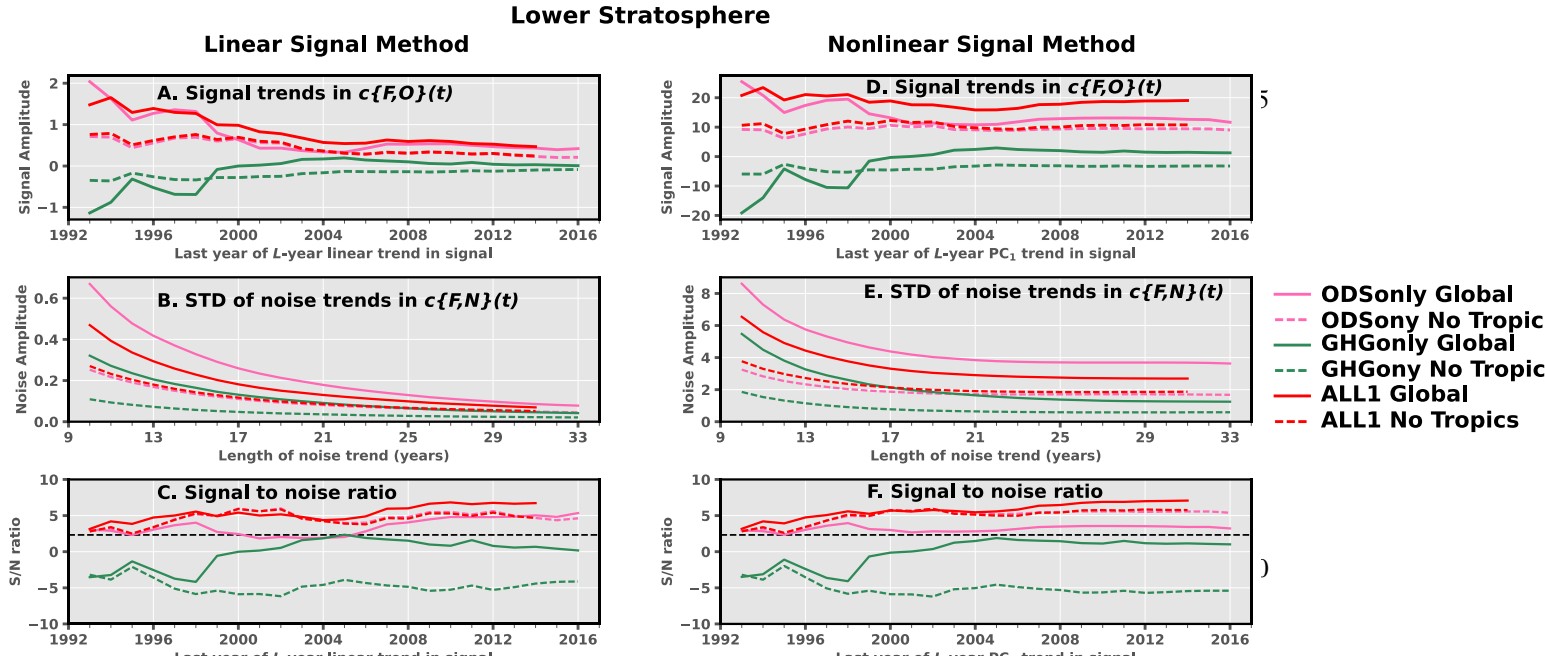

**Figure 8: Same as Fig. 7 but for the lower stratosphere. The solid line shows results for the global lower stratosphere, while the dashed lines exclude the tropics (30ºS to 30ºN).**

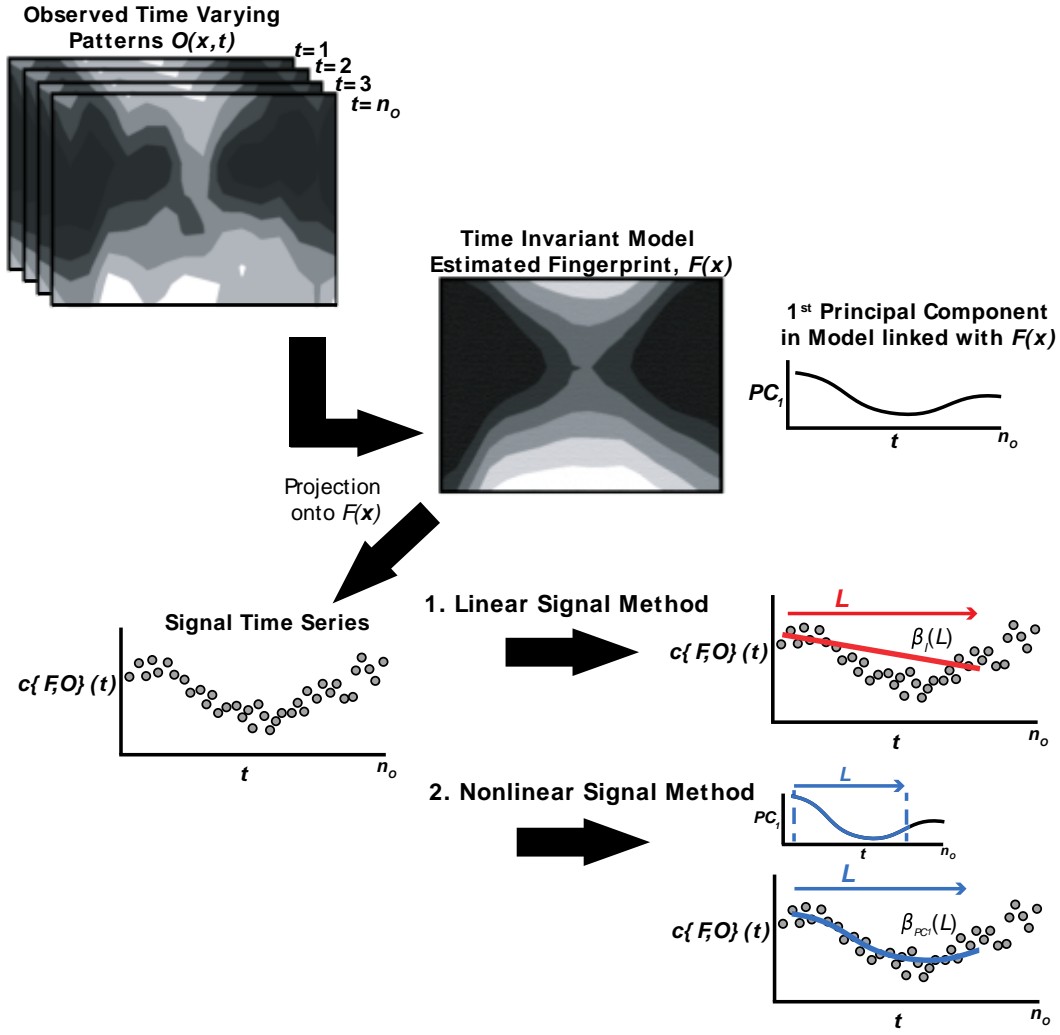

**Figure 9: Schematic of the linear and nonlinear signal methods used for fingerprint identification.** $O(x, t)$ denotes the observed time-varying patterns of zonal-mean ozone anomalies; x is an index over the spatial dimensions (latitude and pressure). $F(x)$ is the searched-for "fingerprint" - the leading empirical orthogonal function of the zonal-mean ozone response in the ODSonly, GHGonly, or ALL1 simulation. For each of the three fingerprints, there is an associated leading principal component time series ($PC_1$) which spans the time period of the observational record. The observations are projected onto each fingerprint, yielding the signal time series, $c\{F, O\}(t)$. In the linear signal method, it is assumed that the time evolution of the fingerprint pattern is quasi-linear over the length of the observational record, and $L$-length linear regression coefficients are calculated, between $c\{F,O\}(t)$ and the time in years; these are the coefficients $\beta_l(L)$. In the nonlinear method, the regression is between $L$-year segments of $c\{F,O\}(t)$ and the model $PC_1$ of the corresponding fingerprint. These are the regression coefficients $\beta_{PC_1}(L)$.

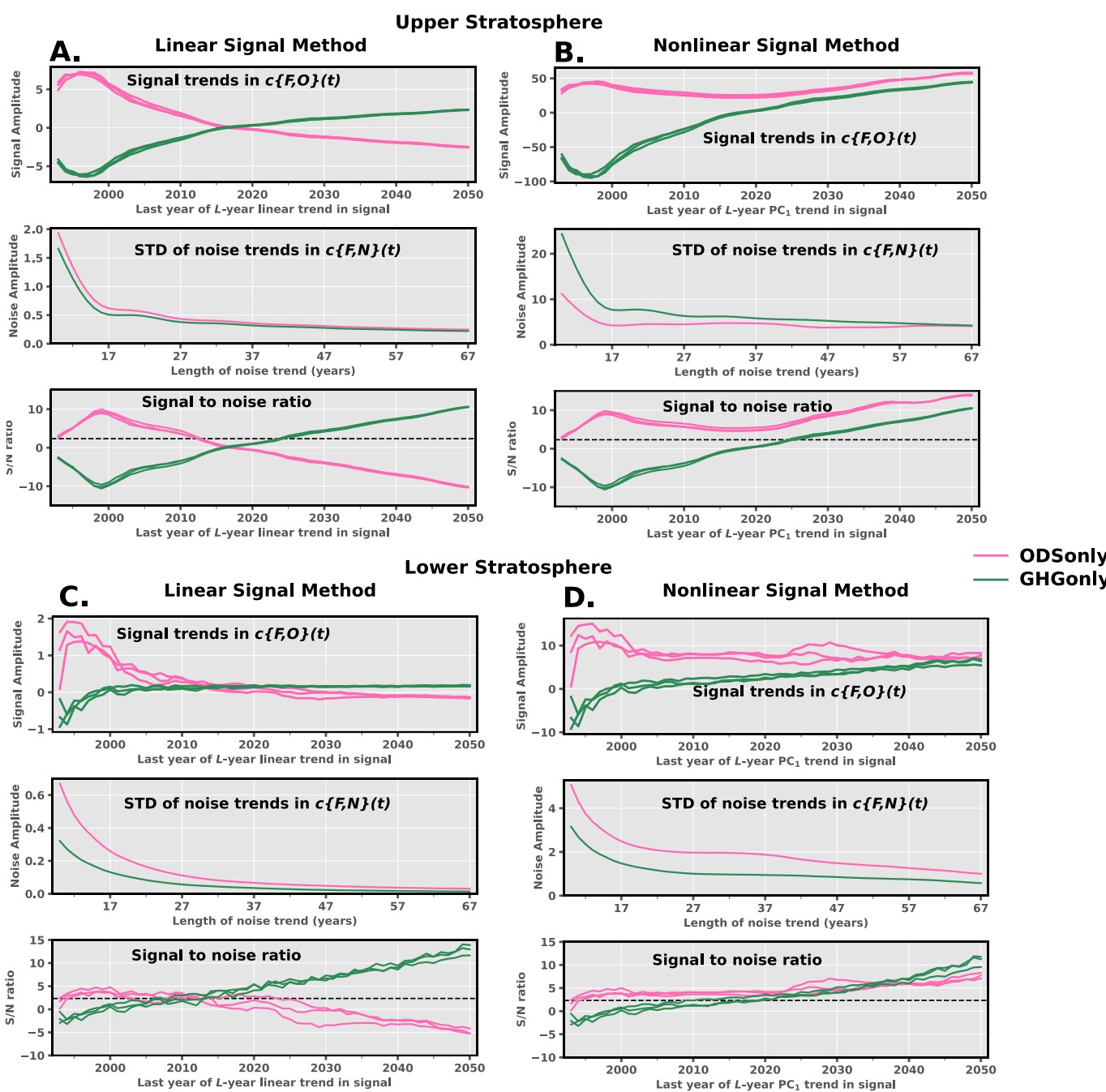

**Figure 10:** Same as in Figs. 7 and 8 but the ALL2 ensemble members (3 total) are used as surrogate observations, yielding information regarding the estimated detection time for the ODSonly and GHGonly fingerprints in model data. The $\beta_l$ and $\beta_{PC_1}$ results in A) and B) are for the upper stratosphere (respectively); results in C) and D) are the corresponding method 1 and method 2 for the lower stratosphere.

| | ALL1 | GHGonly | ODSonly | NAT-h | NAT | CTL |
|---|---|---|---|---|---|---|
| ALL1 | -- | -0.66 | 0.99 | -0.18 | -0.42 | -0.32 |
| GHGonly | 0.19 | -- | -0.72 | 0.18 | -0.10 | 0.19 |
| ODSonly | 0.72 | -0.18 | -- | -0.13 | -0.35 | -0.38 |
| NAT-h | 0.19 | -0.35 | -0.13 | -- | 0.19 | 0.46 |
| NAT | -0.25 | 0.32 | 0.05 | -0.93 | -- | -0.16 |
| CTL | 0.09 | 0.24 | -0.03 | 0.90 | 0.97 | -- |

**Table 1: Correlation coefficients between the leading signal and noise patterns for the upper stratosphere (top right triangle) and lower stratosphere (bottom left triangle). Signal patterns are the leading EOFs for ALL1, GHGonly and ODSonly. The noise patterns are the leading EOFs for NAT-h, NAT and CTL (see Figs. 5 and 6).**

| $F(x,p)$ | Upper stratosphere | | Lower stratosphere | |
|---|---|---|---|---|
| | $\beta_l$ | $\beta_{PC_1}$ | $\beta_l$ | $\beta_{PC_1}$ |
| ALL1 | 4.34 | 9.66 | 6.72 (4.68) | 6.93 (5.82) |
| ODSonly | 3.27 | 7.68 | 5.35 (4.43) | 3.21 (5.42) |
| GHGonly | -3.49 | -3.81 | 0.18 (-4.13) | 0.32 (-4.89) |

**Table 2: Signal-to-noise ratios for the longest $L$-year analysis periods shown in Figs. 7 and 8. Results are for the linear ($\beta_l$) and nonlinear signal methods ($\beta_{PC_1}$). For ODSonly and GHGonly, the longest analysis period is 33 years (1984-2016). For ALL1, the longest analysis period, is 31 years (1984-2014). The figures in brackets for the lower stratosphere indicate the signal-to-noise ratios when the tropics are excluded.**