# Peer review of "Detectability of the Impacts of Ozone Depleting Substances and Greenhouse Gases upon Stratospheric Ozone Accounting for Nonlinearities in Historical Forcings"

_Atmospheric Chemistry and Physics, 2017_

## Referee Comment (RC1) · Anonymous Referee #3 · 11 Aug 2017

The Bandoro et al. manuscript presents a modified approach to detection and attribution that is able to account for the non-linear temporal behaviour of the forcing terms - in the case of stratospheric ozone analysed here, the rise in the atmospheric concentration of ozone depleting substances (ODSs) until the late 1990s followed by a slow decline. The authors clearly and convincingly present a comparison of their new approach with the more widely used approach of assessing signal-to-noise using linear trends and demonstrate the difficulties that arise when the magnitude of the forcing is not linearly changing with time. I have no significant concerns with the methodol-

ogy or the presentation, though I will admit to having little background in detection and attribution.

My one, I believe relative inconsequential, concern is how the NAT-h timeseries was constructed. On Page 7, lines 6 – 8, the authors state:

'In addition, we were able to isolate the response to volcanic aerosols, solar cycle, and the QBO, by differencing the sum of ensemble mean anomalies of the FIXED GHG1960 and ODS1960 simulations from the individual ensemble anomalies of ALL2.'

I can see how the arithmetic of the construction of NAT-h should work; by adding together the effects of GHGs (from the ODS1960 simulations) and ODSs (from the FIXED GHG1960 simulations) and removing these signals from the ALL2 timeseries. My concern is whether the volcanic aerosol effect will be correctly represented in the NAT-h timeseries. The response to large eruptions, such as Pinatubo, will depend critically on the concentrations of reactive chlorine in the stratosphere. Under the low chlorine loading of the ODS1960 simulation the increased volcanic aerosols will produced an increase in ozone in the mid-stratosphere, while under the higher chlorine loading of the FIXED GHG1960 the Pinatubo eruption will produce some increases in the mid-stratosphere and more significant decreases in the lower stratosphere. Given the spatial variability in the response, and regions of the atmosphere where the response to enhanced aerosols in the ODS1960 and FIXED GHG1960 simulations will be of opposite direction, I would think that it would be difficult to imagine that the actual representation of the effects of volcanic aerosols in the NAT-h timeseries would be correct.

I suggest this is probably not a significant concern because the region of the lower stratosphere from 100 to 40 hPa is below the region where volcanic aerosols have the largest impact on the reactive nitrogen chemistry, although it is the region where the opposing responses of halogen and nitrogen chemistry to aerosols is important. And I am not sure how such a relatively rare event as large volcanic eruptions would

contribute to noise in long-term signals. Although I would like to stress that there are reasons to believe the NAT-h simulation is not a good representation of the effects of volcanic aerosols on ozone that one would find in a proper natural historical simulation.

Aside from this footnote to the NAT-h simulation, my other comments are minor and are given below.

The first five paragraphs in the introductory text, over Pages 2 and 3, bounce around a bit from topic to topic, in particular the fourth paragraph (Page 3, Lines 1-9) that discusses linear trends for D&A in the middle of a general discussion of stratospheric ozone. Personally, I found it a bit difficult to follow the thread through the introduction.

Page 6, Lines 17-30 – The term 'emission' is frequently used through this section when I think the more accurate term would be concentration. For example, at Line 27 there is the statement '...while another keeps GHG emissions fixed at their 1960 conditions...' which would suggest the atmospheric concentrations continued to increase after 1960.

Page 6, Line 28 – Do the time-varying concentrations of ODSs in the FIXED GHG1960 simulations affect the radiative forcing of these coupled simulations?

Page 9, Lines Lines 11-18 – Here you argue for conducting the analysis of the lower stratosphere for both the global and extratropical regions. Given the discrepancy with SWOOSH observations for the tropics after 2005, I think this is fully warranted. My point would be about the possible explanation for the discrepancy in tropical ozone being related to the behaviour of volcanic aerosols. If the small volcanoes had significantly impacted ozone and was not properly accounted for in the database of specified aerosols used in the modelling, wouldn't the effect be most pronounced in the extratropical lower stratosphere? Reactive chlorine levels would be much higher than in the tropical lower stratosphere and since many of these small eruptions were in the extratropics I would think the aerosols would also be more prevalent in the extratropical lower stratosphere.

It must also be kept in mind that the agreement of different observational databases for the tropical lower stratosphere is not terribly good. See Section 2.2.4.3 of the WMO 2014 Ozone Science Assessment for the discussion of differences in post-2000 trends for the lower tropical stratosphere.

Page 12, Lines 13-18 – On the underestimation of variability in the upper stratosphere, part of the discrepency may be due to observational uncertainty as different ozone datasets have some significantly different representations of the magnitude of the solar cycle – see Maycock et al., Atmos. Chem. Phys., 16, 10021-10043, 2016. Chemistry-climate models also tend to have solar cycle variations in ozone that are towards the lower end of observational estimates – see Chapter 8.5 of SPARC CCMVal (2010) (SPARC Report on the Evaluation of Chemistry-Climate Models, V. Eyring, T. G. Shepherd, D. W. Waugh (Eds.), SPARC Report No. 5, WCRP-132, WMO/TD-No. 1526).

Page 16, Line 10 – the statement that 'NAT-h is nudged to reanalysis temperature and wind fields.' seems a bit misleading as it makes it sound like a 'Specified Dynamics' simulation where nudging is applied everywhere. The statement should be more specific to the nudging used here to produce the QBO.

Page 17, Line 8 – I am unclear what is meant by '...the noise data set N(x,p,t), which is constructed by concatenating the NAT, NAT-h and CTL simulations.' Does this mean a single timeseries was create by splicing all three of these simulations together, thus creating a ~1250 year timeseries? If so, how would the resulting timeseries be used with the S/N analysis that begins in 1984?

Page 21, Line 18 – there is a erroneous bracket at '.. with methods 1 and 2 (respectively.'

Page 21, Line 28 – there is a word missing at '..method 2 yielded markedly S/N ratios...'

---

## Referee Comment (RC2) · Anonymous Referee #2 · 5 Oct 2017

This paper certainly uses a sledgehammer to crack a nut, I am afraid. It seems to construct a point of ozone trends assumed to be linear, even though the trend is non-linear and by doing so seems to mix two slightly different types of non-linearity.

The first type of linearity (non-linearity) focusses on the temporal behaviour of the ozone time series. However, I do not know of anybody who assumes that the full behaviour of the time series is linear. This is exactly the reason why two carefully chosen periods are fitted with (independent) linear trends, or why people assume a fit to (E)ESC. In addition, non-linear (quadratic) terms have been used to consider early

starting points (pre-1980) of the time series (most recently in Langematz et al., 2016, citing the earlier work as well), clearly acknowledging the complexity of the long-term trend.

The second type of linearity (non-linearity) focusses on the attribution problem. Are the attributed variabilities in ozone a sum of the different terms or not. MLR uses the implicit assumptions that the different factors are a sum, which is presumably a good approximation when different terms in the equation a largely independent of each other. However, it has been realised that this is not always the case and that we have to think carefully of how to choose our proxies (a lot of work is covering this question).

The paper does not clearly separate the two issues and in a way circumnavigates its own problems by choosing two different vertical regimes (nothing wrong with this). However, it would be interesting to know how the method would fair in a more holistic approach.

In summary, I believe the paper to be a nice little exercise in advanced statistics. It is certainly worth publishing after mayor revisions, but the paper needs to simplify its message should clearly acknowledge that the problem of linearity is well recognized (in both aspects – the temporal behaviour and the summing-up of contributing terms). Testing the limits of linear assumptions is always interesting, but it can be done in simple ways with idealised model simulations, alleviating the need for very fancy statistical models. However, I admit that this is a personal preference and that the paper will be a nice contribution to this discussion when revised.

Some more specific comments:

Abstract, line 26: One "the" too many . . .

Page 3, paragraph 1: strange discussion - non-linear versus piecewise linear (fit EESC), see comment above. This discussion and scoping of the paper needs to change most.

Page 4, line 14: I assume you talk about the absolute value, otherwise I suggest "regression coefficient significantly different from zero".

Having the NAT run with no QBO worries me slightly – the authors mention the fact, but I would hope for a slightly more critical assessment of this shortcoming, given that many people try to eliminate the QBO signal in their trend estimates.

You say: "...and there are post-2005 differences between the historical WACCM model simulations and SWOOSH data that are relevant to the interpretation of the D&A results." I certainly agree. However, I would hope for a clearer discussion of what the implications are.

You say: "The decadal variability is of key interest in D&A studies, since it constitutes ..." What indications do we have that the modelled decadal variability is similar to the observed? Many models show distinct attenuations of amplitudes when free running (compared to SD runs). Is this of no concern for WACCM, or are there no sizeable differences for the free running model compared to the SD configuration?

You say: "... simple linear regression line is not an adequate representation of ozone changes over the entire observational record (1984-2016)." As I mentioned above, nobody is stating this (any more, see comment above). Please clarify this.

You use spectral filters to construct a comparison of variability on different time scales. I would prefer simple power spectra comparing the variability. The filtering you do, makes me feel uncomfortable, give that you have one time window up-to 20 years with a time series of ∼33 years.

Figure 3: typo in title

Langematz, U., Schmidt, F., Kunze, M., Bodeker, G. E., and Braesicke, P.: Antarctic ozone depletion between 1960 and 1980 in observations and chemistry–climate model simulations, Atmos. Chem. Phys., 16, 15619-15627, https://doi.org/10.5194/acp-16-15619-2016, 2016.

---

## Author Comment (AC1) · 6 Nov 2017

Response to RC1 and Referee #3:

*Referee #3's comments are shown with a vertical bar next to them, our comments are shown below each of them.*

The Bandoro et al. manuscript presents a modified approach to detection and attribution that is able to account for the non-linear temporal behaviour of the forcing terms - in the case of stratospheric ozone analysed here, the rise in the atmospheric concentration of ozone depleting substances (ODSs) until the late 1990s followed by a slow decline. The authors clearly and convincingly present a comparison of their new approach with the more widely used approach of assessing signal-to-noise using linear trends and demonstrate the difficulties that arise when the magnitude of the forcing is not linearly changing with time. I have no significant concerns with the methodology or the presentation, though I will admit to having little background in detection and attribution.

My one, I believe relative inconsequential, concern is how the NAT-h timeseries was constructed. On Page 7, lines 6 – 8, the authors state:

'In addition, we were able to isolate the response to volcanic aerosols, solar cycle, and the QBO, by differencing the sum of ensemble mean anomalies of the FIXED GHG1960 and ODS1960 simulations from the individual ensemble anomalies of ALL2.'

I can see how the arithmetic of the construction of NAT-h should work; by adding together the effects of GHGs (from the ODS1960 simulations) and ODSs (from the FIXED GHG1960 simulations) and removing these signals from the ALL2 timeseries. My concern is whether the volcanic aerosol effect will be correctly represented in the NAT-h timeseries. The response to large eruptions, such as Pinatubo, will depend critically on the concentrations of reactive chlorine in the stratosphere. Under the low chlorine loading of the ODS1960 simulation the increased volcanic aerosols will produced an increase in ozone in the mid-stratosphere, while under the higher chlorine loading of the FIXED GHG1960 the Pinatubo eruption will produce some increases in the mid-stratosphere and more significant decreases in the lower stratosphere. Given the spatial variability in the response, and regions of the atmosphere where the response to enhanced aerosols in the ODS1960 and FIXED GHG1960 simulations will be of opposite direction, I would think that it would be difficult to imagine that the actual representation of the effects of volcanic aerosols in the NAT-h timeseries would be correct.

I suggest this is probably not a significant concern because the region of the lower stratosphere from 100 to 40 hPa is below the region where volcanic aerosols have the largest impact on the reactive nitrogen chemistry, although it is the region where the opposing responses of halogen and nitrogen chemistry to aerosols is important. And I am not sure how such a relatively rare event as large volcanic eruptions would contribute to noise in long-term signals. Although I would like to stress that there are reasons to believe the NAT-h simulation is not a good representation of the effects of volcanic aerosols on ozone that one would find in a proper natural historical simulation.

Referee #3 raises an important and valid point related to the linear additivity of the stratospheric ozone response to large volcanic eruptions, and the issue of how the NAT-h simulation was constructed. In the low halogen loading stratospheric conditions of FIXED ODS1960, increases in mid-stratospheric ozone following a large volcanic eruption are expected as the loss of ozone at these altitudes is dominated by $NO_x$, due to enhancement of $N_2O_5$ hydrolysis (e.g. Tie and Brasseur, 1995). In the enhanced halogen loading conditions of FIXED GHG1960 and ALL2, the mid-stratospheric increase is limited to higher altitudes, and ozone depletion will occur throughout the lower stratosphere. As Referee #3 correctly points out, NAT-h is constructed by summing the ensemble mean ozone responses in FIXED ODS1960

and FIXED GHG1960, and then by subtracting these responses from the ALL2 realizations. We recognize that the stratospheric ozone response to volcanic aerosols may not be well represented in NAT-h in the vertical region where the reactive nitrogen chemistry is dominant. Our lower stratospheric region (from 40 to 100 hPa) may indeed encompass the region of where ozone concentrations are influenced by the competitive effects of halogen and nitrogen chemistry.

Potential errors in the ozone response to volcanic eruptions in NAT-h are now acknowledged in the revised manuscript. Below, we argue that such errors are unlikely to have substantial impact on our primary D&A results.

To identify slowly-evolving ODS and GHG signals in observational estimates of stratospheric ozone changes, it is critical to have information regarding the natural variability of ozone on timescales of 2-3 decades. It is the background multi-decadal "noise" that is of most interest here. As shown in our Figure 3, NAT-h (which was obtained by subtraction) reliably captures the observed amplitude of the decadal-timescale variability of lower stratospheric ozone arising from external solar forcing. This is the region we are most concerned with in addressing the additivity issue raised by Referee #3. NAT-h also replicates shorter-timescale (< 3 year) observed lower stratospheric ozone variability associated with the QBO and ENSO.

Errors of the form mentioned by Referee #3 (i.e., errors in the lower stratospheric ozone response to short-term, episodic volcanic forcing) are unlikely to affect our estimates of the ODSonly and GHGonly fingerprint patterns, the projections of the SWOOSH ozone data onto these fingerprints, or the estimated variability on multi-decadal timescales from NAT-h. Any errors in the simulated ozone response to volcanic forcing will have largest impact on the short-term variability shown on the x-axis in Fig. 3 – not on the longer multi-decadal variability that is most relevant for the signal-to-noise (S/N) calculations.

Figures 8 E and F help to illustrate the robustness of our results to potential errors in the response of ozone to volcanic forcing. Consider S/N results for the ODS signal in the lower stratosphere in the "tropics excluded" case. This is the case most relevant to the Referee's concern, since the mid-latitudes are the regions where the opposing nitrogen/halogen chemistry would be important. For this particular example, S/N ratios for the full 33-year SWOOSH period are significant at the 5% level or better. To negate these significant results would require that noise trends are roughly 40% larger on the 33-year timescale. It is highly unlikely that an error of this magnitude could be caused by the ozone response to short-term volcanic forcing.

We fully agree that it would be preferable to directly estimate the ozone response to historical volcanic and solar external forcing – i.e., to have access to a simulation in which only volcanic and solar forcing are varied. Unfortunately, such a simulation was not available for our study – which is why we had to estimate the volcanic and solar responses by differencing simulations. We note that such differencing operations are not unique to our study. For example, in Gillett et al., 2011, their ODS response was estimated by differencing of combined forcing simulations and GHG single forcing simulations.

We thank Referee #3 for bringing this concern to our attention. We are confident that our attribution results would not differ significantly if we had access to a more comprehensive historical natural simulation (rather than estimating the volcanic and solar signals by subtraction). Nevertheless, the revised manuscript discusses Referee #3's concern and provides a better description of how NAT-h was actually calculated.

Aside from this footnote to the NAT-h simulation, my other comments are minor and are given below.

- The first five paragraphs in the introductory text, over Pages 2 and 3, bounce around a bit from topic to topic, in particular the fourth paragraph (Page 3, Lines 1-9) that discusses linear trends for D&A in the middle of a general discussion of stratospheric ozone. Personally, I found it a bit difficult to follow the thread through the introduction.

We agree with Referee #3 and have reorganized the introductory text in the revised manuscript to ensure that the discussion of general attribution theory and stratospheric ozone changes are separated.

- Page 6, Lines 17-30 – The term 'emission' is frequently used through this section when I think the more accurate term would be concentration. For example, at Line 27 there is the statement '...while another keeps GHG emissions fixed at their 1960 conditions...' which would suggest the atmospheric concentrations continued to increase after 1960.

Yes Referee #3 is absolutely correct, it should be concentration and not emission. We have changed all such occurrences in the main text.

- Page 6, Line 28 – Do the time-varying concentrations of ODSs in the FIXED GHG1960 simulations affect the radiative forcing of these coupled simulations?

Yes, the time varying concentrations of ODSs in FIXED GHG1960 are radiatively active. It is important, and has been made explicitly clear in the revised manuscript.

- Page 9, Lines Lines 11-18 – Here you argue for conducting the analysis of the lower stratosphere for both the global and extratropical regions. Given the discrepancy with SWOOSH observations for the tropics after 2005, I think this is fully warranted. My point would be about the possible explanation for the discrepancy in tropical ozone being related to the behaviour of volcanic aerosols. If the small volcanoes had significantly impacted ozone and was not properly accounted for in the database of specified aerosols used in the modelling, wouldn't the effect be most pronounced in the extratropical lower stratosphere? Reactive chlorine levels would be much higher than in the tropical lower stratosphere and since many of these small eruptions were in the extratropics I would think the aerosols would also be more prevalent in the extratropical lower stratosphere. It must also be kept in mind that the agreement of different observational databases for the tropical lower stratosphere is not terribly good. See Section 2.2.4.3 of the WMO 2014 Ozone Science Assessment for the discussion of differences in post-2000 trends for the lower tropical stratosphere.

We have taken Referee #3's advice, and substituted our speculation in the revised manuscript about the difference being due to volcanic eruptions (which as Referee #3 points out is not a reasoned hypothesis) with the difference possibly due to errors in the observational data in the tropical lower stratosphere. We thank Referee #3 for offering a more rational explanation.

- Page 12, Lines 13-18 – On the underestimation of variability in the upper stratosphere, part of the discrepency may be due to observational uncertainty as different ozone datasets have some significantly different representations of the magnitude of the solar cycle – see Maycock et al., Atmos. Chem. Phys., 16, 10021-10043, 2016. Chemistry climate models also tend to have solar cycle variations in ozone that are towards the lower end of observational estimates – see Chapter 8.5 of SPARC CCMVal (2010) (SPARC Report on the Evaluation of Chemistry-Climate Models, V. Eyring, T. G. Shepherd, D. W. Waugh (Eds.), SPARC Report No. 5, WCRP-132, WMO/TD-No. 1526).

Referee #3 is correct: we failed to mention that the underestimation of variability may also be due to observational uncertainty, particularly in terms of known deficiencies in the ability of chemistry-climate models to capture ozone variations related to the solar cycle. We have included the reference provided by Referee #3 and added a brief discussion of this point.

- Page 16, Line 10 – the statement that 'NAT-h is nudged to reanalysis temperature and wind fields.' seems a bit misleading as it makes it sound like a 'Specified Dynamics' simulation where nudging is applied everywhere. The statement should be more specific to the nudging used here to produce the QBO.

We agree with Referee #3 that we did not properly describe how the QBO is implemented in WACCM. As with many comparable climate models, the QBO is imposed as an artificial forcing. This is achieved by nudging tropical stratospheric zonal-mean winds to either cyclic or fixed-phase winds, or to the observed winds. We have now revised and improved the description of how the QBO is forced in WACCM.

- Page 17, Line 8 – I am unclear what is meant by '...the noise data set N(x,p,t), which is constructed by concatenating the NAT, NAT-h and CTL simulations.' Does this mean a single timeseries was create by splicing all three of these simulations together, thus creating a ~1250 year timeseries? If so, how would the resulting timeseries be used with the S/N analysis that begins in 1984?

All of the natural and internal climate variability simulations, NAT (1000 years), NAT-h (3x50 years), and CTL (200 years) were concatenated together to form a 1350 year noise data set, N(x,p,t). This was not that clear in the original manuscript. The revised manuscript contains a clarification of this concatenation procedure in Section 6.

- Page 21, Line 18 – there is a erroneous bracket at '.. with methods 1 and 2 (respectively.'

Corrected in text, thank you!

- Page 21, Line 28 – there is a word missing at '..method 2 yielded markedly S/N ratios...'

Corrected in text, thank you!

---

## Author Comment (AC2) · 6 Nov 2017

Response to RC2 and Referee #2:

*Referee #2's comments are shown with a vertical bar next to them, our comments are shown below each of them.*

This paper certainly uses a sledgehammer to crack a nut, I am afraid. It seems to construct a point of ozone trends assumed to be linear, even though the trend is nonlinear and by doing so seems to mix two slightly different types of non-linearity.

The first type of linearity (non-linearity) focusses on the temporal behaviour of the ozone time series. However, I do not know of anybody who assumes that the full behaviour of the time series is linear. This is exactly the reason why two carefully chosen periods are fitted with (independent) linear trends, or why people assume a fit to (E)ESC. In addition, non-linear (quadratic) terms have been used to consider early starting points (pre-1980) of the time series (most recently in Langematz et al., 2016, citing the earlier work as well), clearly acknowledging the complexity of the long-term trend.

The second type of linearity (non-linearity) focusses on the attribution problem. Are the attributed variabilities in ozone a sum of the different terms or not. MLR uses the implicit assumptions that the different factors are a sum, which is presumably a good approximation when different terms in the equation a largely independent of each other. However, it has been realised that this is not always the case and that we have to think carefully of how to choose our proxies (a lot of work is covering this question).

The paper does not clearly separate the two issues and in a way circumnavigates its own problems by choosing two different vertical regimes (nothing wrong with this). However, it would be interesting to know how the method would fair in a more holistic approach.

We appreciate Referee #2's comments above, and hope we can address the issue concerning linearity below. Before addressing specifically, the concerns about the temporal behavior of stratospheric ozone and attribution, we think it would aid to reiterate and clear up any misunderstanding to the detection and attribution (D&A) method used in this study. As addressed on pages 4 and 5 in the original manuscript, there are two established methodologies used in D&A studies:

1) The first one is the optimal regression methodology, which combines the spatial and temporal climate response into a single space-time vector, with the observations being modeled as a linear sum of the simulated responses to individual forcings (see e.g. Allen and Tett, 1999). Each response is scaled by a regression coefficient, expressing the strength of the space-time response pattern in observations. The underlying premise in this methodology is that the observations can be well represented as a linear combination of the input model signal response fields with an additive noise term due to internal variability. Thus, it is assumed that the response patterns to individual forcings are statistically separable and the sum of the responses is equivalent to the response obtained when all forcings are varied together. The important difference between this methodology and the one used in our study and described hereafter, is that it combines the spatial and temporal response into a single space-time vector. The optimal regression methodology was used by the only other formal stratospheric ozone attribution paper (Gillett et al., 2011). In their study, they could not separate the detectability of the ODS and GHG responses of stratospheric ozone change, from 1979-2005, but only the combined response. They hypothesized this was due to intrinsic degeneracy between ODS and GHG space-time patterns. Our study was motivated to use the second methodology for D&A, to investigate if we could confidently detect underline{individually} the ODS and GHG signals in observations, and compare their signal-to-noise ratios. The inherent drawback of the space-time optimal regression is that the method fails to separate the spatial and temporal components. It is difficult, therefore, to "deconstruct" the D&A results, and to determine

whether it is spatial and/or temporal correspondence between the model fingerprint and observations that yields positive identification of the fingerprint.

2) The second D&A methodology does not combine spatial pattern and time evolution into a single vector. Instead, it uses pattern similarity statistics to assess the time evolution of the spatial correspondence between the time-varying observations and time-invariant fingerprints (see e.g., Santer et al., 2003, 2013a,b). Fingerprint patterns are typically estimated from model simulations with individual or combined external forcings. Fingerprints are also compared with model-based estimates on natural variability. The key statistical question is whether the change over time in the spatial correspondence between an individual fingerprint and the observations is greater than the random correspondence between the fingerprint and realization of internal and natural climate variability. The underlying assumption in this approach (which we refer to as method 1) is that the fingerprint pattern does not change markedly as function of time – which we verified for stratospheric ozone by estimating the ODSonly and GHGonly fingerprints over different time periods. In most previous applications of method 1, it has been reasonable to assume that the anthropogenic signal component evolves quasi-linearly (Santer et al., 2003, 2013a,b). This assumption is not justifiable for the ODSonly signal, as Reviewer 2 correctly notes, and as we clearly discuss at multiple points in the text. Our modification of method 1 (which we refer to as method 2) directly addresses the non-linearity in the time evolution of the ODSonly signal. By comparing the signal-to-noise (S/N) ratios for method 1 and method 2, we show that for the non-linear ODSonly signal, there is a the substantial enhancement of S/N in method 2. For the relatively linear GHGonly ozone signal, S/N ratios are very similar in method 1 and method 2.

In our opinion, it is scientifically valuable to compare and contrast the results obtained with purely linear and non-linear representation of signal evolution. We are not aware of other studies that have done this in a D&A context. We are comparing the efficacy of different D&A approaches – not "using a sledgehammer to crack a nut".

Referee #2 states "*The first type of linearity (non-linearity) focusses on the temporal behaviour of the ozone time series. However, I do not know of anybody who assumes that the full behaviour of the time series is linear*". We address the first critique related to fitting linear trends to the entire observational period. Evident in Figure 1 of the manuscript, is that linear regression of stratospheric ozone anomalies from 1984-2016 is not ideal with the SWOOSH, ALL1/2, and ODSonly anomalies. Although this is apparent, in our revised manuscript, we make clear that in order to describe the long-term changes for observations and certain simulations, a linear fit over the entire record would have large errors. However, in the model realm of GHGonly, a linear regression for global upper stratospheric ozone anomalies would indeed be appropriate to describe ozone changes over the entire time period. Indeed, in the cases where a linear fit is clearly inadequate, we could use piece-wise linear trends to describe changes in the depletion and recovery eras or use nonlinear methods such as one Referee #2 points out from Langematz et al., 2016. Many D&A studies assume that, to first order, an anthropogenically forced signal evolves linearly. That is clearly not the case with ozone. It is of interest to compare and contrast the D&A results obtained with a (sub-optimal) linear representation of the ODS signal and a representation capturing the non-linearity of the ODS signal. That is what we do here. Furthermore, unlike the piece-wise linear regression studies referred to by Referee #2, our method 2 does not require any subjective decisions to be made regarding the temporal boundary between the ozone depletion and ozone recovery periods.

In D&A studies, it is advantageous to use the full observational record. A longer observational record enables analysts to better characterize forced signals and unforced noise, and therefore improves the ability to separate an anthropogenically forced signal from internal or "total" natural climate variability. For most climate variables, the amplitude of internally generated variability decreases as the analysis period increases (see, e.g., our Figs. 7,8, and 10 in the manuscript). We therefore prefer to rely on the entire 33-year ozone record rather than on shorter, noisier segments of the record.

This is why in our Figure 4, we contrast linear and nonlinear (EESC) estimated changes 1984-2016, to show that changes associated with specified external forcings depend critically on whether the trends account for nonlinearities. Clearly fitting an EESC curve to the GHGonly simulation is not adequate, just as fitting a line to ODSonly from 1984-2016 – this is key to motivate exactly why we developed the nonlinear method in the attribution methodology. Unlike the trends shown in Figure 4, our nonlinear signal method does not depend on finding arbitrary proxies like EESC, or doing quadratic fits to proxies, as it uses information from the time evolution of the forcing in the model simulations. If the forcing is increasing linearly with time, as in the GHGonly case, there is no difference in detectability (signal-to-noise ratios) between using the traditional linear method to our nonlinear method, as shown in the manuscript. The additional purpose of the linear and EESC trends Fig 4, which Referee #2 takes issue with, is to compare the long-term change patterns with our model-derived fingerprints in Figures 5 and 6. The calculated fingerprints are not based on the regression techniques used in Fig 4., but shows that ODSonly spatial structure is qualitatively similar to the EESC regression, and the GHGonly spatial pattern is qualitatively similar to linear trends.

We agree with Referee #2, and have modified the text to further emphasize that past studies have examined long-term changes in stratospheric ozone and have used numerous methods to quantify such changes. We now cite studies that have used piece-wise linear trends to analyze changes over the ozone depletion and ozone recovery eras. As noted above, however, we seek to estimate the detectability of the ODS and GHG fingerprints over the entire observational record. Use of the entire record is beneficial for signal detection – we find that the noise amplitude is decreased by more than a factor of 2 by increasing the record length from 10 years to 30 years. This decrease in noise amplitude substantially increases our estimated S/N ratios. We have no arbitrary start/end points in our method 2, and we do not use any arbitrary proxies. For the ODSonly signal, we clearly show the value of the nonlinear signal method for D&A analysis relative to the traditional linear signal method.

With regard to the second issue of Referee #2 of the attribution problem of linearity, the referee is entirely correct that if we used a multiple linear regression (MLR) methodology to attribution (#1 above) there could be problems with the assumption that the individual response sum is not always equal to the response where all the forcings are varied together. However, we used methodology #2, so that we can compare the relative detectabilities of the GHG and ODS fingerprint patterns in observations and do not have this problem. Larger S/N ratios of one fingerprint pattern over another indicates that one pattern has a greater expression in observations, and the ratios can be compared and if they are detectable above the 95% confidence level from natural/internal variability. Related to Referee #2's concern; in our method, the spatial fingerprint patterns of ODS and GHG are not orthogonal to each other (linearly independent), as discussed in the manuscript. If two different forcings have identical fingerprints it would indeed be impossible to separate the two using the traditional linear signal method. Though the GHG and ODS do differ in their spatial distributions, even if they are not completely orthogonal, and we take into account the model-estimated time evolution in our attribution. The latter is key to distinguish between ODS and GHG in our signal-to-noise results. If Referee #2 is referencing in their second issue, not the attribution problem of linearity (which is answered above), but the linear additivity of the GHG and ODS signals, we present in Figure R1 global upper and lower stratospheric ozone anomalies to answer the question if the sum of the individual ODSonly and GHGonly ozone signals is equivalent to the ozone signal obtained when ODS and GHG concentrations are varied simultaneously? In WACCM simulations analyzed here, we see that to first order, linearity holds (ODSonly + GHGonly ≈ ALL2 - NAT-h) for both the upper and lower stratosphere.

Referee #2 is correct that we should have recognized more coherently that many past studies have tackled the question of the nonlinear long-term changes in ozone to describe stratospheric changes, and we have added additional text to emphasize this. However, our study, shows precisely how this nonlinearity can be

used to our advantage in D&A work that can be applied to other cases in the climate system. There is a clear partitioning of the "linear additivity" issue and the "non-linear signal evolution" issue in the text of the paper. These issues are not convolved in our discussion. In regards to the comment about using a sledgehammer to crack a nut, this type of formal methodology is necessary if we want to be able to confidently ascertain whether human-caused imprints on the climate system are detectable. In an environment where there is still political debate regarding the reality of the human effect on global climate, this type of research may seem like a 'sledgehammer' but is needed to have sound science. We are using a published, extensively-tested D&A method. We have modified the method – in a relatively straightforward way – to deal with nonlinearity in the evolution of the ODSonly signal.

In summary, I believe the paper to be a nice little exercise in advanced statistics. It is certainly worth publishing after mayor revisions, but the paper needs to simplify its message should clearly acknowledge that the problem of linearity is well recognized (in both aspects – the temporal behaviour and the summing-up of contributing terms). Testing the limits of linear assumptions is always interesting, but it can be done in simple ways with idealised model simulations, alleviating the need for very fancy statistical models. However, I admit that this is a personal preference and that the paper will be a nice contribution to this discussion when revised.

We disagree with Referee #2, we are not using "*very fancy statistical models*". As noted above, we are using a modification of a published, well-tested, and relatively straightforward D&A method. Space-time optimal detection is "*fancy*" – but that is not what we are doing here. The advantage of our method 1 and method 2 is that we preserve information about the spatial structure and time evolution of signal and noise (we don't lump the spatial and temporal information into a single vector). This partitioning of spatial and temporal information is helpful in: 1) separating signal and noise; and 2) understanding the spatial and temporal differences between the ODSonly and GHGonly signals.

Some more specific comments:

- Abstract, line 26: One "the" too many . . .

Corrected, thank you

- Page 3, paragraph 1: strange discussion - non-linear versus piecewise linear (fit EESC), see comment above. This discussion and scoping of the paper needs to change most.

We have modified this section of the discussion related to how past studies have dealt with quantifying long-term changes in stratospheric ozone, including the reference Referee #2 provided.

- Page 4, line 14: I assume you talk about the absolute value, otherwise I suggest "regression coefficient significantly different from zero".

Significantly greater than zero is correct here. A regression coefficient significantly greater than zero indicates a detectable response to the forcing, and one that is close to unity indicates that simulated and observed responses are similar in magnitude. A regression coefficient that is negative, indicates that the observed response is in the opposite direction as simulated, thus for detection in the optimal regression method, the lower limit of regression coefficient has to be greater than zero then the signal is said to be detected (see e.g. Hegerl et al., 1996).

- Having the NAT run with no QBO worries me slightly – the authors mention the fact, but I would hope for a slightly more critical assessment of this shortcoming, given that many people try to eliminate the QBO signal in their trend estimates.

It is a challenge to correctly simulate the QBO in general circulation models. In most models, the equatorial waves responsible for driving the QBO (Kelvin, Rossby gravity, inertial gravity, and mesoscale gravity waves) are not well-represented. As discussed in our paper, the QBO in WACCM is imposed as an artificial forcing. The different ways to force the QBO in WACCM are to nudge the tropical stratospheric zonal-mean winds to either fixed-phase, or observed winds. For NAT-h, the historical simulation with natural forcing only, the QBO phases are nudged to match observations from 1960-2016; after 2016, the QBO has a fixed phase through to 2099. The NAT simulation, which spanned the period from 850-1850, is forced by a historical reconstruction of solar variability, but does not have an imposed QBO. As Referee #2 correctly points out, it would be preferable to have QBO variability in the NAT simulation. We note, however, that: 1) we have no millennial-timescale information on fluctuations in the QBO; and 2) historical changes in the QBO are represented in the NAT-h simulation, which we use for estimating the noise in method 1 and method 2.

- You say: ". . . and there are post-2005 differences between the historical WACCM model simulations and SWOOSH data that are relevant to the interpretation of the D&A results." I certainly agree. However, I would hope for a clearer discussion of what the implications are.

Because of the described differences in the post-2005 behavior of tropical lower stratospheric ozone in ALL1 and SWOOSH, we did partition our S/N analysis for two cases: a global domain and for one poleward of 30S and 30N (excluding the tropics). As discussed in the paper, the ODSonly fingerprint signal was found to be detectable with both inclusion and exclusion of the tropics, with higher S/N ratios if excluded.

Referee #2 is correct that in the conclusion/discussion we did not discuss the implications of the differences between post-2005 lower stratospheric ozone in ALL1 and SWOOSH, it was only discussed Section 3. We have now added such a discussion (see the new Section 7 of revised manuscript). A brief summary of the discussion is that because of the noticeable divergence between simulated and observed post-2005 ozone changes in the tropical lower stratosphere, inclusion of the tropics reduces S/N ratios for our method 2 (i.e., the temporal evolution of ozone differs over for the last decade in observations and in the ODSonly signal, thus reducing the regression coefficient in method 2). We showed that leaving out the tropical lower stratosphere (where the post-2005 divergence between simulations and observations is most pronounced) yields higher S/N ratios in Figure 8 of the manuscript. As pointed out by the other referee for the paper, another important motivation for performing the analysis with both the tropical lower stratosphere included and excluded, is that there is well-known disagreement of different observational databases for the tropical lower stratosphere - as the much lower abundances of ozone in that region lead to larger instrumental uncertainties. The WMO 2014 report discusses differences in post-2000 trends for the lower tropical stratosphere.

- You say: "The decadal variability is of key interest in D&A studies, since it constitutes . . ." What indications do we have that the modelled decadal variability is similar to the observed? Many models show distinct attenuations of amplitudes when free running (compared to SD runs). Is this of no concern for WACCM, or are there no sizeable differences for the free running model compared to the SD configuration?

A detailed paper on the comparison of WACCM's climate and its variability can be found in Marsh et al., 2013. Referee #2 is correct that our D&A study relies on a single climate model, and we acknowledged the shortcomings of a "single model" analysis in the introduction and discussion. Model bias would affect

our results. To quantify the bias, and to ensure that the decadal variability in stratospheric ozone is close to observations, the original manuscript explicitly includes a section (3.2) that carried out such an analysis, so we refer back to that section.

- You say: ". . . simple linear regression line is not an adequate representation of ozone changes over the entire observational record (1984-2016)." As I mentioned above, nobody is stating this (any more, see comment above). Please clarify this.

Yes, as discussed earlier this has been clarified, as this was to motivate the development of our nonlinear signal method in the attribution section.

- You use spectral filters to construct a comparison of variability on different time scales. I would prefer simple power spectra comparing the variability. The filtering you do, makes me feel uncomfortable, give that you have one time window up-to 20 years with a time series of ~33 years.

The band-pass filter focuses on variability on the timescale of 10 years with half-power points at 5 and 20 years. Figure R2, included below, shows the response functions for the Butterworth band-pass and high-pass filters. This is the same filtering approach used by Santer et al., 2011, who examined variability of the temperature of the lower troposphere in model simulations and observations.

In our original submission, we included power spectra in the supplement Figure S2. As expected, the NAT simulation has peaks around 11 years for the solar cycle in the upper stratosphere, and NAT-h has peaks in the 2 to 3 year range related to the QBO.

- Figure 3: typo in title

This has been fixed, thank you!

Langematz, U., Schmidt, F., Kunze, M., Bodeker, G. E., and Braesicke, P.: Antarctic ozone depletion between 1960 and 1980 in observations and chemistry–climate model simulations, Atmos. Chem. Phys., 16, 15619-15627, https://doi.org/10.5194/acp-16- 15619-2016, 2016.

Response Figures:

[Figure]

Figure R1: Global ozone anomalies in the upper and lower stratosphere, similar to Figure 1 in the manuscript, but here we subtract the NAT_h response from ALL_2 to investigate the agreement between GHGonly+ODSonly and ALL2 - NAT_h.

[Figure]

Figure R2: Response functions for Butterworth high-pass and band-pass filters used in the main text.

Response References:

Allen, M. R. and Tett, S. F. B.: Checking for model consistency in optimal fingerprinting, Clim. Dyn., 15(6), 419–434, doi:10.1007/s003820050291, 1999.

Gillett, N. P., Akiyoshi, H., Bekki, S., Braesicke, P., Eyring, V., Garcia, R., Karpechko, A. Y., McLinden, C. A., Morgenstern, O., Plummer, D. A., Pyle, J. A., Rozanov, E., Scinocca, J. and Shibata, K.: Attribution of observed changes in stratospheric ozone and temperature, Atmos. Chem. Phys., 11(2), 599–609, doi:10.5194/acp-11-599-2011, 2011.

Hegerl, G. C., von Storch, H. , Hasselmann, K., Santer, B.D., Cubasch, U., and Jones, P.D.: Detecting anthropogenic climate change with an optimal fingerprint method. J. Climate, 9, 2281–2306, 1996.

Marsh, D. R., Mills, M. J., Kinnison, D. E., Lamarque, J.-F., Calvo, N. and Polvani, L. M.: Climate Change from 1850 to 2005 Simulated in CESM1(WACCM), J. Clim., 26(19), 7372–7391, doi:10.1175/JCLI-D-12-00558.1, 2013.

Santer, B., Wigley, T. and Jones, P.: Correlation methods in fingerprint detection studies, Clim. Dyn., 1993.

Santer, B.D., Taylor, K.E., Wigley, T.M.L., Penner, J.E., Jones, P.D., and Cubasch, U.: Towards the detection and attribution of an anthropogenic effect on climate, Climate Dynamics 12(77). https://doi.org/10.1007/BF00223722, 1995.

Santer, B. D., Mears, C., Doutriaux, C., Caldwell, P., Gleckler, P. J., Wigley, T. M. L., Solomon, S., Gillett, N. P., Ivanova, D., Karl, T. R., Lanzante, J. R., Meehl, G. A., Stott, P. A., Taylor, K. E., Thorne, P. W., Wehner, M. F. and Wentz, F. J.: Separating signal and noise in atmospheric temperature changes: The importance of timescale, J. Geophys. Res. Atmos., 15 116(D22), doi:10.1029/2011JD016263, 2011.

Santer, B. B. D., Painter, J. F. J., Mears, C. a, Doutriaux, C., Caldwell, P., Arblaster, J. M., Cameron-Smith, P. J., Gillett, N. P., Gleckler, P. J., Lanzante, J., Perlwitz, J., Solomon, S., Stott, P. a, Taylor, K. E., Terray, L., Thorne, P. W., Wehner, M. F., Wentz, F. J., Wigley, T. M. L., Wilcox, L. J. and Zou, C.-Z.: Identifying human influences on atmospheric temperature., Proc. Natl. Acad. Sci. U.S.A., 110(1), 26–33, doi:10.1073/pnas.1210514109, 2013a.

Santer, B. D., Painter, J. F., Bonfils, C., Mears, C. A., Solomon, S., Wigley, T. M. L., Gleckler, P. J., Schmidt, G. A., Doutriaux, C., Gillett, N. P., Taylor, K. E., Thorne, P. W. and Wentz, F. J.: Human and natural influences on the changing thermal structure of the atmosphere., Proc. Natl. Acad. Sci. U. S. A., 110(43), 6–11, doi:10.1073/pnas.1305332110, 2013b.